# CANONICAL TREE COVER NEURAL NETWORKS FOR EXPRESSIVE AND INVARIANT GRAPH LEARNING

**Michael Ito**
University of Michigan
mbito@umich.edu

**Danai Koutra**
University of Michigan
dkoutra@umich.edu

**Jenna Wiens**
University of Michigan
wiensj@umich.edu

## ABSTRACT

While message-passing NNs (MPNNs) are naturally invariant on graphs, they are fundamentally limited in expressive power, oversmooth, and oversquash. Canonicalization offers a powerful alternative by mapping each graph to a unique, invariant representation on which expressive non-invariant encoders can operate. However, existing approaches rely on a single canonical sequence that distorts graph distances and restricts expressivity. To address these limitations, we introduce *Canonical Tree Cover Neural Networks* (CTNNs), which represent the graph with a canonical spanning tree cover. Each tree is then processed with an expressive tree encoder. Theoretically, tree covers better preserve graph distances in comparison to sequences, and on sparse graphs, the cover recovers all edges with a logarithmic number of trees in the graph size, making CTNNs strictly more expressive than sequence-based canonicalization approaches. Empirically, CTNNs consistently outperform invariant GNNs and sequence-based canonical GNNs across sparse molecular and protein graph classification benchmarks. Overall, CTNNs advance graph learning by providing an efficient, invariant, and expressive representation learning framework on sparse graphs via tree cover-based canonicalization.

## 1 INTRODUCTION

In graph representation learning, capturing a graph's natural symmetries (i.e., isomorphism invariance) is essential for learning and generalization. One way to enforce this invariance is to bake it directly into the architecture: message-passing neural networks (MPNNs) (Duvenaud et al., 2015; Gilmer et al., 2017; Kipf and Welling, 2017) achieve architectural invariance by iteratively aggregating neighbor embeddings, but are provably equivalent in expressive power to the 1-dimensional Weisfeiler–Leman test (Xu et al., 2019; Morris et al., 2019), and suffer from oversmoothing (Li et al., 2018; Chen et al., 2020) and oversquashing (Oono and Suzuki, 2020; Di Giovanni et al., 2024), and are thus fundamentally limited. A second approach achieves invariance via random sampling: random walk neural networks (RWNNs) (Tönshoff et al., 2023; Chen et al., 2025; Kim et al., 2025; Ito et al., 2025) sample walks as input to powerful sequence models, overcoming MPNN limitations but incurring potentially prohibitive sampling costs when training on large datasets. A complementary line of work relies on canonicalization, which maps each graph to a unique representative, allowing any expressive, non-invariant model to operate on a fixed, invariant input, bypassing expensive sampling (Bloem-Reddy and Teh, 2020; Kaba et al., 2023). In this work, we establish the limitations of existing canonicalization approaches on graphs and propose a new canonicalization.

Existing graph canonicalization approaches first assign labels to each node, flatten the graph into a single sequence, either via learned sorting layers (Niepert et al., 2016; Zhang et al., 2018; Grover et al., 2019) or through traversal as in canonical SMILES (Goh et al., 2017; Honda et al., 2019), and then feed the sequence into a powerful downstream sequence model. In this work, we formally quantify how flattening into a sequence distorts graph distance. To illustrate this limitation, consider $S_n$, the $n$-node star (Figure 1, $n = 7$). Each leaf node in the graph has distance 1 to the center node, while leaf nodes in the sequence necessarily have distance $O(n)$ to

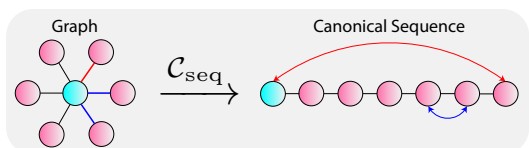

Figure 1: Canonical sequence representations introduce significant stretch and contraction.

the center node (stretch). Moreover, while leaves have distance 2 to each other in $S_n$, certain leaves have distance 1 in the sequence (contraction). Thus, the sequence-based canonicalization can stretch and contract original distances, making structure harder to capture. We further establish that the reduction of the graph into a single sequence limits the expressivity of the overall approach to that of its node labeler, discarding the benefits of using powerful downstream models.

To address these limitations, we propose *Canonical Tree Cover Neural Networks* (CTNNs), which construct a canonical spanning tree cover via minimum spanning tree extraction and coverage-aware edge label refinement. Each tree in the cover is processed by an existing expressive tree encoder (Tai et al., 2015) and aggregating over the cover yields an invariant representation. Importantly, CTNNs leverage coverage-aware edge labelers that are initialized by a node labeler: when using a canonical graph node labeler that assigns unique labels to all nodes (e.g., NAUTY (McKay and Piperno, 2014)), the resulting tree cover is fully deterministic and invariant; when using inexpensive, structurally meaningful labelers (e.g., degree, centrality, or 1-WL), tie-breaking introduces randomness, leading to probabilistic invariance while preserving useful inductive biases. By leveraging tree representations and capturing structure across a set of canonical representatives, CTNNs better capture graph distances and are more expressive than sequence-based canonical GNNs. Across sparse molecular and protein graph classification tasks, CTNNs consistently outperform architecturally invariant GNNs and existing canonical GNNs. In summary, we make the following contributions:

- **Current Limitations of Canonical GNNs.** We establish that sequence-based graph canonicalization methods fail to preserve graph distance and are limited in expressivity.
- **New Canonical GNN: Canonical Tree Cover Neural Networks (CTNNs).** We introduce CTNNs, which construct a canonical tree cover. Each tree is then processed by expressive recurrent tree encoders and aggregated to obtain an invariant representation.
- **Theory: Invariance, Distance Preservation, and Expressivity Guarantees.** We prove that CTNNs produce invariant graph representations, preserve graph distance information, and exceed the expressivity of sequence-based canonical GNNs and MPNNs. With universal tree encoders, CTNNs achieve universality on invariant graph functions.
- **Extensive Empirical Evaluation.** Across molecular and protein graph classification benchmarks, CTNNs outperform architecturally invariant GNNs and canonical baselines.

## 2 BACKGROUND AND PRELIMINARIES

We first introduce notation and review canonical approaches on graphs, the primary family of models under investigation. These approaches typically produce a single sequence that is fed to a sequence model. We then formalize recurrent sequence models, which often outperform attention and convolution on graphs by better matching the traversal inductive bias. Despite their practical performance, however, recurrent sequence models can suffer from long graph-derived sequences. These limitations lead us to consider recurrent tree models that instead propagate information along trees, which we will later demonstrate better capture graph distance.

### 2.1 NOTATION ON GRAPHS AND TREES

Let $G = (V, E, \mathbf{X})$ be an undirected graph with $n = |V|$ nodes, $m = |E|$ edges, and node features $\mathbf{X} \in \mathbb{R}^{n \times d}$. For $v \in V$, let $\mathbf{x}_v$ denote the $v$-th row of $\mathbf{X}$, $\mathcal{N}(v) = \{u \in V : (u, v) \in E\}$ its neighborhood, and $\deg(v) = |\mathcal{N}(v)|$ and $d_G(u, v)$ the shortest path distance in $G$. A rooted tree is $T = (V, E, \mathbf{X}, r)$ with root $r \in V$. Each non-root node $v \neq r$ has a unique parent $p(v)$, and we write $C(v) = \{u \in V : p(u) = v\}$ for its children. Leaf nodes of the tree satisfy $C(v) = \varnothing$.

### 2.2 MESSAGE-PASSING GRAPH NEURAL NETWORKS AND GNN EXPRESSIVITY

Message-passing GNNs (MPNNs) update node representations by pooling representations from local neighborhoods via a permutation-invariant aggregator (Duvenaud et al., 2015; Gilmer et al., 2017; Kipf and Welling, 2017). Concretely, for a graph $G$ and node $i \in V$, one layer takes the form

$$f_{\text{MPNN}}(G)_i = f_{\text{agg}}(\{\mathbf{x}_j : j \in \hat{\mathcal{N}}(i)\}),$$

where $\hat{\mathcal{N}}(i)$ denotes the self-loop augmented neighborhood of $i$ and $f_{\text{agg}}$ is permutation-invariant. While this invariance is desirable, it also limits distinguishability, and MPNNs cannot separate certain

families non-isomorphic graphs (Xu et al., 2019; Azizian and Lelarge, 2021). To compare model expressivity, we use the standard preorder on graph-level maps: for two GNNs $f_1, f_2$, define

$$f_2 \preceq f_1 \iff \forall G, H : f_1(G) = f_1(H) \Rightarrow f_2(G) = f_2(H).$$

Equivalently, $f_1$ is at least as discriminative as $f_2$. We write $f_2 \prec f_1$ if $f_2 \preceq f_1$ and the inclusion is strict (i.e., some $G, H$ are separated by $f_1$ but not by $f_2$), and $f_1 \simeq f_2$ if both $f_2 \preceq f_1$ and $f_1 \preceq f_2$ hold. This ordering aligns with approximation power: if $f_2 \prec f_1$, then any target representable by $f_2$ is representable by $f_1$, while the converse fails in general.

## 2.3 CANONICAL GRAPH NEURAL NETWORKS

Graph canonicalization aims to obtain a unique isomorphism–invariant node labeling (McKay, 1981). Because computing an exact canonical labeling is as hard as the graph isomorphism problem, practical methods adopt soft approximations (e.g., GNN embeddings). After obtaining an approximate labeling, these pipelines typically flatten the graph into a single sequence either via sorting layers (Niepert et al., 2016; Zhang et al., 2018) or through traversal such as canonical SMILES (Goh et al., 2017; Honda et al., 2019; Chithrananda et al., 2020), allowing expressive sequence models to process the sequence. Formally, let $\pi_V : V \to \mathbb{R}$ be a node labeling function (e.g., MPNN), $\mathcal{C}_{\text{seq}}$ be a single-sequence canonicalizer that maps the labeled graph $(G, \pi_V)$ to a sequence depending only on $\pi_V$ and carrying only the node features $\mathbf{X}$, and $f_{\text{seq}}$ be a sequence model. A general sequence–based canonical GNN is defined as

$$f_{\text{CanSeq}}(G) = f_{\text{seq}}(\mathcal{C}_{\text{seq}}(G, \pi_V)).$$

As a concrete instance, if $\pi_V$ is an MPNN, $\mathcal{C}_{\text{seq}}$ is a differentiable sorting layer, and $f_{\text{seq}}$ is a 1D CNN, then $f_{\text{CanSeq}}$ recovers Deep Graph Convolutional Neural Network (Zhang et al., 2018).

## 2.4 RECURRENT SEQUENCE AND TREE MODELS

Recent RWNNs find that recurrence often outperforms attention and convolution by better matching the traversal inductive bias (Wang and Cho, 2024; Chen et al., 2025; Ito et al., 2025). Given inputs $(\mathbf{x}_t)_{t=1}^T$, initial state $\mathbf{h}_0$, and state transition map $\Phi : \mathbb{R}^d \times \mathbb{R}^d \to \mathbb{R}^d$, the recurrent update is defined

$$\mathbf{h}_t = \Phi(\mathbf{h}_{t-1}, \mathbf{x}_t), \qquad \text{for } t = 1, \ldots, T.$$

Recurrent models suffer on long sequences that exacerbate vanishing/exploding gradients, which motivates our use of recurrent tree models that shorten dependency paths and mitigate these instabilities. Recurrent tree models generalize sequence recurrence to rooted trees (Tai et al., 2015; Xiao et al., 2024), propagating information bottom–up from children to their parent. Given $T = (V, E, r)$ with $L$ levels and node inputs $\{\mathbf{x}_v\}_{v \in V}$, recurrent tree models compute hidden states $\{\mathbf{h}_v\}_{v \in V}$ by applying a local transition to child states and aggregating with a permutation–invariant operator $f_{\text{agg}}$:

$$\mathbf{h}_v = f_{\text{agg}}(\{\Phi(\mathbf{h}_c, \mathbf{x}_v) \mid c \in C(v)\}) \quad \text{for } \ell = L, \ldots, 0 \text{ and all } v \text{ with } d_T(v, r) = \ell,$$

with $f_{\text{agg}}(\varnothing) = 0$ for leaves. Setting $\Phi(\mathbf{h}_c, \mathbf{x}_v)$ as a standard LSTM update recovers the Tree LSTM of Tai et al., 2015. The tree representation is taken as $\mathbf{h}_r$ at the root. In Section 4, we propose a canonicalization of graphs via spanning tree covers that can be used as input to recurrent tree models.

## 3 LIMITATIONS OF SEQUENCE-BASED CANONICALIZATIONS

In this section, we characterize the limitations of single-sequence canonical GNNs. First, we quantify how sequence canonicalization distorts graph structure, stretching and contracting graph distances. We next turn to expressivity and demonstrate that even when the sequence model is universal, the canonical GNN is no more expressive than its node labeler because it relies on a single canonical representative. Together, these limitations motivate our tree cover–based canonicalization, which better preserves distances and increases expressivity by operating on a cover of spanning trees.

### 3.1 DISTANCE DISTORTION UNDER SEQUENCE CANONICALIZATION

To formalize how sequence canonicalization fails to preserve structure, we use distortion (Matoušek, 2013), which quantifies the stretch/contraction in distance after mapping points between spaces. Intuitively, we prefer canonicalizations with lower distortion, better preserving the original distances.

Figure 3: Sequence canonicalization is only as expressive as its labeler $\pi_V$ despite using a universal downstream sequence model. $f_{\text{CanSeq}}$ thus fails to distinguish graphs $\pi_V$ fails to distinguish.

**Definition 3.1** (Distortion). Let $(X, d_X)$ and $(Y, d_Y)$ be metric spaces. A mapping $f : (X, d_X) \to (Y, d_Y)$ has distortion $D \geq 1$ if there exists $r > 0$ such that for all $x, y \in X$,

$$r\, d_X(x, y) \ \leq \ d_Y\big(f(x), f(y)\big) \ \leq \ D\, r\, d_X(x, y).$$

Let $(G, d_G)$ denote a graph $G$ with shortest–path distance $d_G$, and let $\mathcal{C}_{\text{seq}}(G, \pi_V)$ be its single canonical sequence under $\pi_V$. Equip $\mathcal{C}_{\text{seq}}$ with the distance $d_{\text{seq}}(u, v) = |\sigma(u) - \sigma(v)|$, where $\sigma : V \to \{1, \ldots, |V|\}$ is the induced ordering. Let $\varphi(G)$ be the *graph bandwidth* (Díaz et al., 2002), which measures the smallest maximum stretch over any edge when $G$ is laid out on a line:

$$\varphi(G) \ = \ \min_{\sigma} \ \max_{(u,v) \in E} \ |\sigma(u) - \sigma(v)|.$$

Existing work studies embeddings $f : V \to \mathbb{R}$ into the real line and imposes a non-contractive constraint, under which the distortion reduces to max stretch and is lower bounded by graph bandwidth (Heggernes et al., 2011; Dragan et al., 2014). In our setting, however, we consider permutations $\sigma : V \to \{1, \ldots, n\}$, which are the natural inputs to sequence models and can exhibit both stretch and contraction. Under permutation embeddings, we provide an analogous result.

**Corollary 3.2** (Graph bandwidth lower bounds sequence distortion). *Let $D_{\text{seq}}$ be the distortion of $\mathcal{C}_{\text{seq}}(G, \pi_V)$ from $(G, d_G)$ to the line with distance $d_{\text{seq}}$. Then, for any $\pi_V$, $\varphi(G) \ \leq \ D_{\text{seq}}$.*

All proofs are in Appendix A. The bandwidth lower bound gives a concrete, well-studied graph metric to evaluate the distortion of $\mathcal{C}_{\text{seq}}$. Although $\varphi(G)$ is hard to compute in general, it is known for many families (Figures 1, 2): on $n$-node stars $S_n$ and cliques $K_n$ one has $\varphi(S_n) = \varphi(K_n) = \Theta(n)$, so any single–sequence canonicalization incurs the worst-case linear distortion; on complete binary trees $\varphi(T_{2,\ell}) = \Theta(2^\ell/\ell) = \Theta(n/\log n)$, and on cycles $C_n$ and paths $P_n$ one has $\varphi(P_n) = \varphi(C_n) = \Theta(1)$. Beyond specific families, the bound offers general insights. Given that $\varphi(G) \geq (n-1)/\text{diam}(G)$,

$D_{\text{seq}}$ is at least $(n-1)/\text{diam}(G)$. It is also monotone under edge addition, indicating that highly connected graphs, reflected by larger algebraic connectivity $\lambda_2$, force larger distortion. These effects negatively impact the sequence model: distorted distances make structure more difficult to capture. Importantly, any method relying on sequences, including canonicalizations and sampling approaches like RWNNs, inherits these limitations. **To address the limitations of sequences, we turn to tree representations.**

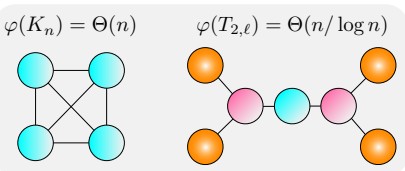

Figure 2: $\varphi(G)$ for $n$-node clique, $K_n$, and complete binary tree with $\ell$ levels, $T_{2,\ell}$.

## 3.2 Expressive Limitations of Sequence-based Canonical GNNs

Beyond the limitations of sequence representations due to distortion, we characterize the expressive limitations of the canonical GNN due to relying only on a single sequence. Formally, we show that $f_{\text{CanSeq}}$ when equipped with universal $f_{\text{Seq}}$ is only as expressive as its node labeler $\pi_V$.

**Proposition 3.3** ($\pi_V$ and $f_{\text{CanSeq}}$ are equally expressive). *Let $f_{\text{CanSeq}}$ be a canonical sequence–based model with universal $f_{\text{seq}}$ and let $\pi_V$ be its labeling function. Then, $f_{\text{CanSeq}} \simeq \pi_V$.*

If $\pi_V$ is an MPNN, its power matches 1-WL; consequently, $f_{\text{CanSeq}}$ inherits 1-WL limitations and fails on the same graph families (Figure 3). Crucially, this holds even when $f_{\text{seq}}$ is universal: once information is lost at the labeling stage, no downstream single-sequence canonicalization can recover it, limiting the expressivity of the full pipeline. Although multiple labelers and sequences could be used to improve expressivity, sequences remain limited by distortion. **Thus, in order to address the limitations of the single labeler and sequence, we instead consider multiple labelers and trees.**

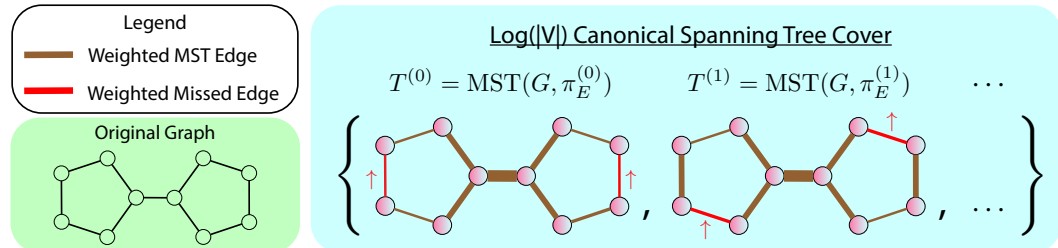

Figure 4: Canonical spanning-tree cover. At iteration $k$, compute $\mathrm{MST}(G, \pi_E^{(k)})$ using coverage-aware edge weights (thicker = larger magnitude weight). Edges missed in $k$ (red) are up-weighted to bias their inclusion in $k+1$. On sparse graphs, the union of $O(\log |V|)$ trees covers all edges.

## 4 CANONICAL TREE COVER NEURAL NETWORKS (CTNNs)

To address the representation limitations of sequences due to distortion and the expressive limitations due to a single sequence, we introduce Canonical Tree Cover Neural Networks (CTNNs), which construct a canonical spanning tree cover. In Section 5, we demonstrate that tree representations better preserve graph distances in comparison to sequences, while a set of canonical trees allows for complete graph reconstruction and is strictly more expressive than a single sequence.

### 4.1 CANONICAL SPANNING TREE COVERS

To construct a canonical spanning tree cover, we leverage coverage-aware edge labelers and minimum spanning tree (MST) samplers rather than a fixed node labeler and sequence canonicalizer. By updating edge weights across rounds, later trees are biased toward edges not yet selected, yielding provable coverage across the union of sampled trees. Formally, let $G$ be a graph and at iteration $k \in \{0, \ldots, K-1\}$ for hyperparameter $K$ let $\pi_E^{(k)} : E \to \mathbb{R}$ be an edge labeler. Let $\mathcal{C}_{\mathrm{tree}}$ be an MST extractor that maps an edge–labeled graph $(G, \pi_E^{(k)})$ to a spanning tree $T^{(k)}$ according to weights $\pi_E^{(k)}$, setting the root node as the center of $T^{(k)}$. To promote edge coverage across the set, we update the weights by penalizing edges used in the last tree $T^{(k)}$ with hyperparameter $\tau$. We initialize with any isomorphism-invariant node labeler $\pi_V$ (e.g., degree), which biases MSTs towards edges incident to high label nodes. Formally, the update and initialization can be written:

$$\pi_E^{(0)}(e) = -\big(\pi_V(e_u) + \pi_V(e_v)\big), \; \pi_E^{(k+1)}(e) = \pi_E^{(k)}(e) + \tau \, \mathbb{1}\{e \in T^{(k)}\}$$

We refer to further implementation and pseudocode details of the construction in Appendix B.

### 4.2 INVARIANT CANONICAL TREE NEURAL NETWORKS

Given a canonical cover of MSTs, $\mathcal{T} = \{T^{(k)}\}_{k=0}^{K-1}$, we process each tree with a recurrent tree encoder and augment it with message passing over the remaining non–tree edges to capture the local connectivity missed by each individual spanning tree. Let the residual graph be $G \backslash T^{(k)} := (V, E \setminus E(T^{(k)}))$ and denote $f_{\mathrm{tree}}$ as a recurrent tree encoder (e.g., Tree LSTM (Tai et al., 2015)) and $f_{\mathrm{MPNN}}$ an MPNN (e.g., GIN (Xu et al., 2019)). For each $k \in \{0, \ldots, K-1\}$ and node $i \in V$,

$$f_{\mathrm{TreeMPNN}}\big(T^{(k)}\big)_i = f_{\mathrm{tree}}\big(T^{(k)}\big)_i + f_{\mathrm{MPNN}}\big(G \backslash T^{(k)}\big)_i$$

We then aggregate across the set of trees with a permutation–invariant operator $f_{\mathrm{agg}}$ to obtain

$$f_{\mathrm{CTNN}}(G) := f_{\mathrm{agg}}\Big( \big\{ f_{\mathrm{TreeMPNN}}\big(T^{(k)}\big) : T^{(k)} = \mathcal{C}_{\mathrm{tree}}\big(G, \pi_E^{(k)}\big), \; k = 0, \ldots, K-1 \big\} \Big).$$

**Probabilistic invariance.** When CTNNs use an inexpensive, structurally meaningful node labeler that does not uniquely distinguish vertices (e.g., degree), we obtain probabilistic invariance (Bloem-Reddy and Teh, 2020; Kim et al., 2025; Ito et al., 2025). Such labelers are isomorphism-invariant but may assign identical scores to nodes, so we resolve ties using random tie-breaking. This induces an isomorphism-invariant distribution over spanning tree covers. Formally, for any permutation $g \in \mathbb{S}_n$ acting on $G$ by relabeling nodes, the random output $f_{\mathrm{CTNN}}(G)$ has the same distribution

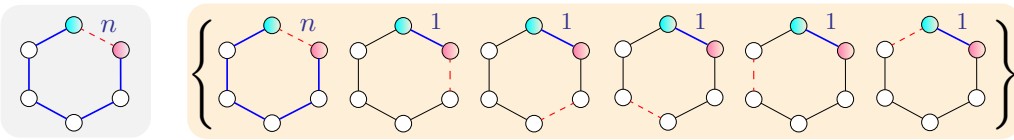

Figure 5: Single tree distortion is $O(n)$ on $C_n$, while expected distortion is constant over a spanning tree distribution since on average the distance between any two nodes is small.

as $f_{\text{CTNN}}(g \cdot G)$. Consequently, the averaged predictor $\mathbb{E}\big[f_{\text{CTNN}}(G)\big]$ is an invariant function on graphs. In this regime, CTNN relies on a small amount of randomness to break symmetries, but that randomness is controlled by the underlying canonicalization (i.e., node labeler).

**Theorem 4.1** (Probabilistic invariance of CTNNs). *A randomized graph representation $X(G)$ is probabilistically invariant if its distribution is unchanged under any node relabeling, i.e., $X(G) \stackrel{d}{=} X(g \cdot G)$ for every permutation $g \in \mathbb{S}_n$. The random output $f_{\text{CTNN}}(G)$ is probabilistically invariant:*

$$f_{\text{CTNN}}(G) \stackrel{d}{=} f_{\text{CTNN}}(g \cdot G) \quad \text{for all } g \in \mathbb{S}_n.$$

*Then, $\Phi(G) := \mathbb{E}\big[f_{\text{CTNN}}(G)\big]$ is an invariant function satisfying $\Phi(G) = \Phi(g \cdot G)$ for all $g \in \mathbb{S}_n$.*

**Deterministic invariance.** At the other end of the spectrum, one can instantiate CTNN with a true graph canonicalization tool such as NAUTY (McKay and Piperno, 2014), which computes a canonical labeling that separates all nodes up to isomorphism. With such a canonical node labeler and an injective initialization of edge weights the induced tree cover becomes a deterministic canonical representation: isomorphic graphs are mapped to exactly the same set of trees, and $f_{\text{CTNN}}(G) = f_{\text{CTNN}}(g \cdot G)$ holds for all permutations $g$. This is particularly beneficial when the graph exhibits a high degree of symmetry such as complete or regular graphs, where weaker node labelers result in many ties. CTNN thus provides a unified framework that interpolates between fully deterministic canonicalization and probabilistic invariance, depending on the choice of node labeler.

### 4.3 RUNTIME COMPLEXITY

CTNN preprocessing is primarily dominated by constructing the $K$ MSTs and cost of $\pi_V$. Using Kruskal's algorithm (Kruskal, 1956), the total cost is $O(K m \log n + \pi_V)$, which is efficient on sparse graphs where $m = O(n)$ and for inexpensive $\pi_V$ (e.g., degree). A major practical advantage of canonicalization is that these trees are computed once before training and reused across epochs, eliminating on-the-fly sampling incurred by sampling approaches. The computation parallelizes naturally across graphs, and the memory cost is small ($O(Kn)$ edges per graph). Empirically, we show this preprocessing time is efficient across datasets (Appendix E.2).

## 5 DISTORTION AND EXPRESSIVITY BOUNDS FOR CTNNS

We first analyze distance preservation: because CTNNs aggregate over spanning trees, they yield distortion bounds that better preserve graph distance in comparison to single-sequence canonicalization. We then turn to expressivity, establishing the benefits of sets of canonical representatives. On sparse graphs our tree cover recovers the full edge set with only $O(\log m)$ trees, which has two immediate consequences for expressivity: (i) CTNNs are strictly more expressive than single–sequence canonicalizations, and (ii) when paired with universal tree encoders, CTNNs become universal.

### 5.1 EXPECTED DISTORTION BOUNDS FOR CTNNS

We first analyze how well CTNNs preserve distances, establishing distortion bounds for $\mathcal{C}_{\text{tree}}$. Because CTNNs sample MSTs, we use probabilistic distortion (Fakcharoenphol et al., 2003).

**Definition 5.1** (Expected distortion). Let $(X, d_X)$ be a metric space and let $\mu$ be a distribution on metrics $\mathcal{M}(X)$. The *expected distortion* of $\mu$ is the least $D \geq 1$ s.t for $r > 0$ and for all $x, y \in X$,

$$r \, d_X(x, y) \; \leq \; \mathbb{E}_{\rho \sim \mu}\big[\rho(x, y)\big] \; \leq \; D \, r \, d_X(x, y).$$

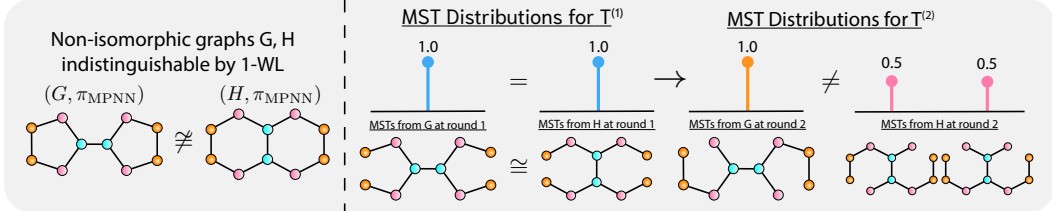

Figure 6: Two non-isomorphic stably colored graphs indistinguishable to 1-WL (left). The first extracted MST $T^{(1)}$ for $G$ is isomorphic to $T^{(1)}$ from $H$ and the distributions are equal, but after reweighting and extracting $T^{(2)}$, the MST distributions diverge, illustrating that multiple canonical trees can be more informative than a single canonical tree (right).

As a baseline, we analyze distortion for uniform spanning trees (USTs). In this regime, the expected tree distance between nodes $u$ and $v$ is upper bounded by their hitting times, the expected number of steps a random walk takes to travel from $u$ to $v$. Empirically, we verify that CTNN tree distributions inherit and can improve upon the low-distortion behavior established by USTs (Appendix E.3):

**Theorem 5.2** (UST expected distortion). *Let $G$ be a graph, and let $T$ be a uniform random spanning tree of $G$. Denote by $H(u,v)$ the random walk hitting time from $u$ to $v$. Then,*

$$D_{\text{UST}} = \max_{u,v} \; \mathbb{E}\big[d_T(u,v)\big]/d_G(u,v)\,, \quad \mathbb{E}\big[d_T(u,v)\big] \;\leq\; H(u,v) + H(v,u)/2.$$

In contrast to the bandwidth lower bound for single–sequence canonicalization, which can force worst-case distortion, the expected UST distortion aligns with random walk distance and preserves structure significantly better on sparse families. Every tree admits a unique spanning tree, so on trees $D_{\text{UST}} = 1$. By comparison, $\mathcal{C}_{\text{seq}}$ incurs distortion $\Theta(n/\log n)$ on balanced trees and $\Theta(n)$ on stars. On $C_n$, distortion is also constant, highlighting the benefit of averaging over trees (Figure 5). Our bounds also provide general insights: tree distances behave well in sparse graphs, where the square root of hitting times and shortest paths scale comparably. In highly dense graphs, however, shortest paths are smaller than hitting times and distortion worsens. Overall, CTNNs yield expected distortion that is small on many sparse structures where in comparison single sequences stretch distances, better capturing graph structure for downstream encoders.

## 5.2 COVERAGE AND EXPRESSIVITY GUARANTEES VIA MST CANONICALIZATION

We now turn to the expressive benefits of CTNNs. Instead of relying on a single canonical representative, CTNNs build a spanning tree cover, providing downstream encoders access to full structure. We first show our coverage–aware MST scheme needs only logarithmically many trees to cover all edges on sparse graphs. We then leverage coverage to show CTNN expressivity is strictly greater than sequence–based canonical GNNs and establish its universality on graph functions.

**Lemma 5.3** (Logarithmic spanning–tree cover). *Let $G = (V, E)$ be a graph with $m = |E|$ and arboricity $\Upsilon(G)$, the minimum number of forests required to cover $G$. Fix any node labeler $\pi_V$ with $\tau \;>\; \max_e \pi_E^{(0)}(e) - \min_e \pi_E^{(0)}(e)$. Denote $\mathcal{T} = \{T^{(k)}\}_{k=0}^{K-1}$ as the set of trees produced by a CTNN. If $K \geq \Upsilon(G) \ln m$ iterations, the union of the MSTs covers all edges: $\bigcup_{k=0}^{K-1} E\big(T^{(k)}\big) \;=\; E$.*

Importantly, on sparse graphs, arboricity is constant, and CTNNs obtain full coverage with $K \geq O(\log(|V|))$. As established in Section 3, $f_{\text{CanSeq}}$ is only as expressive as $\pi_V$. CTNNs, by contrast, operate on a tree cover enabled by coverage-aware edge label refinement, and as a result, are *strictly* more expressive than their initial labeler. In this setting, $f_{\text{CanTree}}$ is a randomized function due to the randomness induced by $\pi_V$. Hence, we use the following probabilistic notion of distinguishability.

**Proposition 5.4** ($f_{\text{MPNN}} \prec f_{\text{CanTree}}$). *Let $f_{\text{CanTree}}$ be a CTNN satisfying Lemma 5.3, equipped with $\pi_V \simeq f_{\text{MPNN}}$. Then, $\forall G, H$, $f_{\text{CanTree}}(G) \stackrel{d}{=} f_{\text{CanTree}}(H) \implies f_{\text{MPNN}}(G) = f_{\text{MPNN}}(H)$. Moreover, $\exists G \not\cong H$ such that $f_{\text{CanTree}}(G) \stackrel{d}{\neq} f_{\text{CanTree}}(H)$ while $f_{\text{MPNN}}(G) = f_{\text{MPNN}}(H)$.*

In cases where the graphs exhibit a high degree of symmetry, and $\pi_V$ fails to distinguish them initially, the reweighting scheme and tree cover allow CTNNs to distinguish the graphs, exceeding the expressivity of $\pi_V$ (Figure 6). When CTNNs are equipped with a canonical node labeler $\pi_V$ that separates all nodes, universal tree encoder, and full coverage, they are universal.

**Theorem 5.5** (CTNN Universality). *Let $\mathcal{G}$ be a finite class of graphs. Assume: (i) $K$ satisfies Lemma 5.3, (ii) the tree encoder $f_{\text{tree}}$ and aggregation $f_{\text{agg}}$ are universal on their domains, and (iii) $\pi_V$ is a canonical node labeler that separates all nodes. Then for any continuous invariant graph function $f : \mathcal{G} \to \mathbb{R}$ and any $\varepsilon > 0$, there exists a CTNN such that*

$$\sup_{G \in \mathcal{G}} \big| f_{\text{CTNN}}(G) - f(G) \big| \leq \varepsilon.$$

## 6 EXPERIMENTS AND RESULTS

Through empirical evaluation, we aim to answer the following research questions, extending our theory by testing CTNNs on datasets with factors not explicitly addressed in the theoretical analysis (e.g., class imbalance), and including domain–specific canonicalizations beyond our theory, such as molecular fingerprints (Rogers and Hahn, 2010) commonly used in molecular analysis.

- **RQ1 (Discriminative performance).** How does CTNN compare to (i) invariant GNNs upper bounded by 1-WL expressivity (**GCN**, **GAT**, **GIN**), (ii) expressive GNNs more powerful than 1-WL (**GT**, **RWSE**, **GSN**, **ESAN**) and (iii) canonicalization baselines?
- **RQ2 (Distance distortion).** Do CTNNs reduce metric distortion relative to sequence-based canonicalizations, and does this reduction translate into improved task performance?
- **RQ3 (Ablations and sensitivity).** Which components of CTNN contribute most to performance, and how sensitive is performance to their settings?

### 6.1 EXPERIMENTAL SETUP

**Datasets.** We evaluate on molecular and protein benchmarks, domains where canonicalization is widely adopted and frequently used in practice (Goh et al., 2017; Alley et al., 2019) and where long–range dependencies and high expressivity are critical (Dwivedi et al., 2022a). For molecules, we use **ClinTox**, **BACE**, **BBBP**, **HIV**, and **PCBA** datasets from MoleculeNet (Wu et al., 2018). For proteins, we adopt ProteinShake (Kucera et al., 2023) datasets: **SCOP**, **GO MOL**, and **GO BIO**. These tasks span diverse molecule and protein tasks such as molecular activity and protein structure classification. Notably, proteins are larger than molecules, making structure more difficult to capture. To demonstrate CTNNs are applicable to domains in which canonicalization is not yet widely adopted, we evaluate on a larger and denser brain graph classification benchmark (Said et al., 2023), where the task is to predict 1 of 7 mental states (e.g., emotion processing) (Appendix E.4).

**Baselines.** We consider invariant GNNs upper bounded by 1-WL expressivity: (1) **GCN** (Kipf and Welling, 2017), (2) **GAT** (Veličković et al., 2018), (3) **GIN** (Xu et al., 2019). We next consider expressive GNNs strictly more powerful than 1-WL message passing: (4) **GT** (Dwivedi and Bresson, 2021), graph transformers, (5) **RWSE** (Dwivedi et al., 2022b) and (6) **GSN** (Bouritsas et al., 2022), which augment message-passing with additional structural features, and (7) **ESAN** (Bevilacqua et al., 2022), which decompose the graph into subgraphs, processing each component with an MPNN. We also evaluate canonicalization approaches: (8) **Fingerprint** (Rogers and Hahn, 2010), stacking an MLP on hand-crafted chemical descriptors, (9) **SMILES** (Goh et al., 2017), applying sequence models over canonical SMILES, (10) **Primary Seq.** (Alley et al., 2019), applying sequence models to the primary sequence, (11) **DGCNN** (Zhang et al., 2018), a representative sequence-based canonical approach leveraging MPNNs as $\pi_V$ and sorting as $\mathcal{C}_{\text{seq}}$, and (12) **RCM** (Diamant et al., 2023), applying sequence models to the ordering determined by the Cuthill-McKee algorithm. We provide a summary of the design space for all canonicalizations in Appendix C.

**Training and Evaluation.** For all benchmarks, we set $f_{\text{tree}}$ as a Tree-LSTM, $f_{\text{MPNN}}$ as a GIN, $f_{\text{agg}}$ as SUM, $\pi_V(v) = \deg(v)$, and $\tau = 1$. For molecular datasets, we set $K = 4$, and for proteins, we use $K = 8$. Following each dataset's protocol, performance is computed as AUC or accuracy. We report median (min, max) performance over five random splits (60/20/20), which is more robust than mean and standard deviation for small sample sizes. We compute stretch as $\max_{i,j}\{d_{\text{emb}}(i,j)/d_G(i,j)\}$ and contraction as $\max_{i,j}\{d_G(i,j)/d_{\text{emb}}(i,j)\}$. For sequence canonicalizations, $d_{\text{emb}} = d_{\text{seq}}$. For CTNNs, we report expected distortion as the average across the trees (i.e., $\max_{i,j} \text{mean}_k\{d_{T^{(k)}}(i,j)/d_G(i,j)\}$ for stretch). We provide remaining details in Appendix D[1].

---

[1] Code can be found at: https://github.com/MLD3/CanonicalTreeNNs

Table 1: Median (min, max) of model performance ($\times 100$) across test splits. We highlight in **blue** the best model. "NA" indicates not applicable; "OOT" denotes training exceeds the time limit (24h).

| | | Small Molecular Benchmarks | | | Large Molecular Benchmarks | | Protein Benchmarks | | |
|---|---|---|---|---|---|---|---|---|---|
| | | **ClinTox** | **BACE** | **BBBP** | **HIV** | **PCBA** | **SCOP** | **GO BIO** | **GO MOL** |
| | # Graphs | 1.5K | 1.5K | 2K | 41K | 440K | 10K | 22K | 32K |
| | Avg. $|V|$ | 26.1 | 34.1 | 23.9 | 25.5 | 26.0 | 217.5 | 254.5 | 250.1 |
| | Avg. $|E|$ | 28.0 | 36.9 | 26.0 | 27.5 | 28.1 | 593.8 | 698.5 | 687.5 |
| | Metric | AUC ↑ | AUC ↑ | AUC ↑ | AUC ↑ | AUC ↑ | ACC ↑ | AUC ↑ | AUC ↑ |
| MPNN/GT | GCN | 62.4 (56.9, 74.7) | 59.2 (53.9, 64.3) | 73.9 (68.9, 81.4) | 60.1 (57.6, 60.5) | 78.5 (78.0, 79.3) | 63.4 (62.8, 64.9) | 59.2 (57.9, 69.7) | 60.6 (49.8, 84.5) |
| | GAT | 62.1 (55.8, 65.9) | 60.8 (52.0, 75.1) | 77.5 (74.1, 82.8) | 66.3 (63.5, 71.0) | 78.5 (77.2, 79.2) | 58.9 (51.6, 59.9) | 57.0 (53.2, 58.7) | 57.6 (50.3, 81.1) |
| | GIN | 59.7 (54.1, 72.4) | 59.9 (51.4, 71.8) | 75.3 (49.4, 85.3) | 47.7 (45.9, 62.6) | 80.4 (80.1, 81.2) | 68.0 (67.9, 69.2) | 66.3 (59.9, 79.0) | 83.7 (81.5, 85.6) |
| | GT | 57.1 (46.5, 73.5) | 67.1 (57.6, 75.7) | 75.8 (62.6, 84.0) | 69.4 (67.5, 70.6) | 80.6 (79.6, 80.9) | OOT | OOT | OOT |
| | RWSE | 63.6 (56.4, 74.6) | 57.7 (42.3, 69.6) | 73.7 (69.7, 86.5) | 59.4 (54.4, 68.0) | 83.5 (82.5, 83.7) | **74.5 (72.1, 75.5)** | 74.0 (69.8, 75.0) | 85.8 (85.0, 86.1) |
| | GSN | 63.7 (55.6, 68.2) | 70.1 (64.9, 79.1) | 71.4 (64.3, 79.7) | 54.2 (46.3, 69.8) | 83.1 (82.6, 83.4) | **74.5 (73.4, 76.7)** | 71.2 (59.0, 77.5) | 85.0 (76.6, 85.3) |
| | ESAN | 61.8 (56.7, 66.7) | 55.8 (52.3, 69.2) | 74.9 (70.5, 80.6) | 71.9 (55.9, 73.0) | 83.7 (83.5, 84.3) | 66.6 (66.5, 68.5) | 79.6 (78.3, 81.5) | 85.7 (85.6, 86.4) |
| Canonical | Fingerprint | 66.5 (52.3, 74.9) | **82.9 (78.7, 87.8)** | **86.2 (83.4, 92.5)** | **76.2 (73.1, 81.6)** | 84.7 (84.6, 85.2) | NA | NA | NA |
| | SMILES | 62.5 (45.7, 68.6) | 76.5 (68.4, 80.3) | 71.9 (65.5, 75.3) | 65.8 (62.9, 68.0) | 80.4 (80.1, 80.7) | NA | NA | NA |
| | Primary Seq. | NA | NA | NA | NA | NA | 63.0 (60.8, 63.5) | 74.3 (69.2, 79.5) | 85.2 (84.5, 85.8) |
| | DGCNN | 60.1 (27.6, 69.6) | 67.2 (64.1, 74.8) | 75.0 (42.8, 86.4) | 66.6 (61.5, 67.9) | 84.9 (84.0, 85.3) | 65.3 (64.6, 67.8) | 62.0 (59.7, 68.9) | 84.5 (84.0, 84.7) |
| | RCM | 70.7 (48.6, 87.0) | 76.3 (73.3, 81.2) | 84.3 (75.1, 89.1) | **75.5 (73.8, 83.7)** | 84.6 (84.5, 84.9) | 57.0 (56.5, 57.8) | 68.4 (66.7, 69.3) | 83.3 (82.6, 83.7) |
| | CTNN (ours) | **84.7 (78.5, 91.0)** | 79.3 (75.4, 85.0) | 86.1 (80.6, 90.4) | 75.2 (70.3, 83.3) | **87.4 (87.0, 87.5)** | 72.1 (70.5, 74.0) | **82.0 (81.2, 83.2)** | **86.6 (86.4, 87.1)** |

Table 2: Mean $\pm$ s.d. of empirical stretch and contraction across 50 random samples for canonicalizations. In comparison to all canonicalizations, CTNNs significantly reduce stretch and contraction.

| | | Max Stretch ↓ | | | | | | | |
|---|---|---|---|---|---|---|---|---|---|
| | | **ClinTox** | **BACE** | **BBBP** | **HIV** | **PCBA** | **SCOP** | **GO BIO** | **GO MOL** |
| Canonical | SMILES | 17.62 ± 11.91 | 24.34 ± 8.33 | 15.22 ± 8.29 | 20.02 ± 11.31 | 18.82 ± 7.15 | NA | NA | NA |
| | Primary Seq. | NA | NA | NA | NA | NA | 172.6 ± 34.11 | 165.72 ± 44.51 | 173.08 ± 37.83 |
| | DGCNN | 18.2 ± 9.10 | 23.92 ± 6.91 | 14.96 ± 6.09 | 19.16 ± 10.33 | 19.04 ± 5.35 | 196.44 ± 16.03 | 192.56 ± 15.54 | 192.84 ± 14.62 |
| | RCM | 3.92 ± 1.65 | 4.48 ± 0.75 | 3.76 ± 1.06 | 4.36 ± 1.41 | 3.76 ± 0.97 | 34.68 ± 7.68 | 33.76 ± 9.11 | 33.44 ± 8.71 |
| | CTNN (ours) | **2.36 ± 0.90** | **2.40 ± 0.44** | **2.37 ± 0.47** | **2.51 ± 0.56** | **2.29 ± 0.27** | **17.85 ± 3.15** | **17.56 ± 4.56** | **18.12 ± 4.45** |
| | | Max Contraction ↓ | | | | | | | |
| Canonical | SMILES | 4.88 ± 3.16 | 5.32 ± 2.45 | 4.12 ± 2.57 | 4.70 ± 2.07 | 5.26 ± 2.01 | NA | NA | NA |
| | Primary Seq. | NA | NA | NA | NA | NA | 2.72 ± 2.86 | 4.44 ± 5.62 | 5.44 ± 6.31 |
| | DGCNN | 11.34 ± 5.25 | 12.78 ± 2.98 | 9.68 ± 3.00 | 11.74 ± 5.14 | 12.36 ± 3.03 | 16.32 ± 3.25 | 16.04 ± 4.12 | 16.16 ± 4.15 |
| | RCM | 5.66 ± 2.74 | 6.90 ± 2.65 | 4.92 ± 2.09 | 6.20 ± 3.00 | 5.66 ± 2.23 | 12.56 ± 2.04 | 12.16 ± 2.37 | 11.92 ± 2.34 |
| | CTNN (ours) | **1.00 ± 0.00** | **1.00 ± 0.00** | **1.00 ± 0.00** | **1.00 ± 0.00** | **1.00 ± 0.00** | **1.00 ± 0.00** | **1.00 ± 0.00** | **1.00 ± 0.00** |

## 6.2 RQ1 & RQ2: Discriminative Performance and Distance Distortion

CTNNs significantly outperform architecturally invariant MPNNs and GTs, consistent with the theoretical expressivity gains established in Section 5.2 (Table 1). Subgraph GNNs (**RWSE**, **GSN**, **ESAN**) are strong baselines and are particularly competitive on protein datasets, but CTNN exceeds their performance on molecular benchmarks. We attribute this to the fact that, although subgraph GNNs increase theoretical expressivity beyond 1-WL, they still rely on message-passing and inherit known limitations such as oversmoothing and oversquashing. CTNN mitigates these issues by operating on low-distortion spanning tree covers with powerful recurrent tree encoders.

While some canonicalizations are competitive, they depend on domain knowledge and lack generality (e.g., **Fingerprint**). CTNNs outperform or match all sequence-based canonicalizations, including those that are domain-driven and provide one-to-one encodings of their graphs (**SMILES**, **Primary Seq.**), allowing for maximal expressivity. We attribute CTNNs' gains to distortion (Table 2). Across molecular and protein benchmarks, CTNN tree covers achieve substantially smaller stretch than sequence-based canonicalizations. Crucially, trees never contract distances, obtaining optimal contraction = 1. In contrast, sequences exhibit both large stretch and nontrivial contraction. A noteworthy case is **RCM**: its ordering reduces bandwidth and lowers stretch on molecular graphs, yet it still doesn't reach CTNN performance because it incurs contraction. Moreover, on larger protein graphs its stretch increases, underscoring a fundamental limitation of single sequence canonicalization. Collectively, these results align with our theory: canonical spanning-tree covers preserve graph distances significantly better than sequences, enabling stronger downstream models.

On a larger and denser brain graph classification benchmark from NeuroGraph (Said et al., 2023), CTNN also obtains the best performance in comparison to MPNNs (**GIN**) and canonicalization baselines (**RCM** and **DGCNN**) (Appendix E.4). These results demonstrate that CTNN is applicable beyond sparse biochemical domains, where canonicalization is not yet widely adopted.

Table 3: Median (max-min) performance for ablations on benchmarks across test splits. CTNN (full) obtains or matches the best performance across all datasets, supporting each design choice.

| Ablation | Molecular Benchmarks | | | | | Protein Benchmarks | | |
|---|---|---|---|---|---|---|---|---|
| | ClinTox | BACE | BBBP | HIV | PCBA | SCOP | GO BIO | GO MOL |
| Single tree vs. cover | 78.2 (72.2, 88.9) | 78.7 (75.5, 82.1) | 87.4 (80.9, 91.3) | 76.5 (71.3, 81.6) | 87.0 (86.0, 87.5) | 68.7 (67.3, 70.0) | 69.4 (68.5, 70.3) | 79.9 (78.2, 81.7) |
| MPNN vs. TreeMPNN | 82.6 (73.3, 90.0) | 78.2 (71.9, 86.1) | 81.3 (72.4, 85.3) | 73.8 (69.3, 77.1) | 84.9 (84.5, 85.3) | 57.1 (56.3, 58.3) | 78.4 (78.0, 79.2) | 83.9 (83.6, 84.0) |
| TreeRNN vs. TreeMPNN | 84.3 (80.1, 92.0) | 83.5 (80.5, 86.7) | 86.8 (82.6, 92.0) | 77.9 (74.5, 82.4) | 87.0 (86.7, 87.4) | 64.8 (62.5, 65.1) | 78.7 (77.3, 79.1) | 85.3 (84.4, 85.7) |
| CTNN (full) | 84.7 (78.5, 91.0) | 79.3 (75.4, 85.0) | 86.1 (80.6, 90.4) | 75.2 (70.3, 83.3) | 87.4 (87.0, 87.5) | **72.1** (70.5, 74.0) | **82.0** (81.2, 83.2) | **86.6** (86.4, 87.1) |

## 6.3 RQ3: Ablations and Sensitivity

**Ablations.** We evaluate three CTNN variants to isolate which design choices matter most in which settings, including (i) replacing the cover with a single tree, (ii) replacing the TreeMPNN with an MPNN, and (iii) replacing the TreeMPNN with a TreeRNN (Table 3). On molecular benchmarks, the ablations are often competitive with the full model, suggesting CTNN is robust in very sparse tree-like regimes. In contrast, on protein benchmarks the differences are more significant, highlighting when each component becomes important. (i) Using a single canonical tree instead of a cover reduces edge coverage and increases distortion. This has limited impact on sparse molecules, where graph structures are tree-like, but it underperforms on proteins, where additional trees substantially improve coverage and distance preservation on average. (ii) Using an MPNN instead of TreeMPNN processes trees with MPNNs and reintroduces message-passing inductive biases, contributing to oversmoothing and oversquashing, which leads to a consistent drop across most benchmarks, including molecular benchmarks. (iii) Using a TreeRNN instead of TreeMPNN removes the MPNN on residual edges and has little effect on molecules, where few edges remain after MST extraction, but can degrade performance on denser proteins, where residual edges carry meaningful local connectivity. Overall, CTNN (full) matches or achieves the best performance across datasets. The performance gaps across ablations are small on molecular graphs, but increase on protein graphs, indicating that the cover, tree encoder, and residual-edge message passing are primarily important in larger and denser regimes.

**Sensitivity.** We also conduct sensitivity analyses for different choices of number of trees, $K$, node labeler, $\pi_V$, and penalty, $\tau$ (Appendix E.1). Increasing $K$ yields consistent gains on proteins: edge coverage rises rapidly, average distortion decreases, and performance improves. These results align with our theory that only a small number of trees is needed for full coverage on sparse graphs and additional trees better capture original graph distances on average, resulting in increased performance for larger $K$. CTNN is also robust to node labeler $\pi_V$: degree, closeness centrality, and BLISS (Junttila and Kaski, 2007), a canonical node labeler, are each close in performance. In the main experiments, we default to degree for its efficiency. Lastly, CTNN is also stable across the penalty $\tau$, where coverage, distortion, and accuracy follow similar trends across choices of $\tau$.

## 7 Conclusion

In this work, we developed the first theoretical analysis of sequence-based canonicalization for graphs, establishing that sequences distort distances and that single-representative approaches are constrained by the expressivity of their labelers. This analysis covered canonicalizations widely used in practice such as domain-driven sequences including SMILES (Goh et al., 2017; Honda et al., 2019) and primary protein sequences (Alley et al., 2019; Rao et al., 2019), learnable orderings based on GNNs and differentiable sorting (Niepert et al., 2016; Zhang et al., 2018), and algorithmic orderings that optimize bandwidth (Cuthill and McKee, 1969; Diamant et al., 2023). Motivated by this analysis, we introduced *Canonical Tree Cover Neural Networks*, which construct canonical spanning-tree covers and leverage expressive tree encoders. CTNNs are provably invariant, better preserve graph distances, and are more expressive than sequence-based canonical GNNs. Empirically, CTNNs outperform invariant GNNs and canonicalization baselines on molecular and protein benchmarks.

Our coverage and expressivity guarantees rely on sparsity assumptions, and thus, characterizing CTNNs in dense regimes remains open. Despite the focused scope, CTNNs consistently maintain advantages in our experiments, highlighting the value of spanning-tree covers over sequences. More broadly, our results underscore the importance of canonical representations that respect underlying graph geometry. By leveraging canonical tree covers, CTNNs offer an expressive, invariant, and efficient framework for learning on sparse graphs.

ACKNOWLEDGEMENTS

This material is based upon work supported by the U.S. Department of Energy, Office of Science, Office of Advanced Scientific Computing Research, Department of Energy Computational Science Graduate Fellowship under Award Number DE-SC0023112. It was also partially supported by National Science Foundation under Grants No. IIS 2212143 and IIS 2504090, and CAREER Grant No. IIS 184549. We thank the anonymous reviewers and members of the MLD3 and GEMS labs for their valuable feedback.

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

## A    MATHEMATICAL PROOFS

### A.1    DISTANCE DISTORTION UNDER SEQUENCE CANONICALIZATION

**Corollary A.1** (Bandwidth lower-bounds sequence distortion). *Let $G = (V, E)$ be a connected, unweighted graph with shortest-path metric $d_G$. Let $\varphi(G) := \min_\sigma \max_{\{u,v\} \in E} |\sigma(u) - \sigma(v)|$ be the bandwidth of $G$. Then for every ordering $\pi$,*

$$\varphi(G) \;\leq\; D_{\text{seq}}(\pi).$$

*Proof.* For an injective ordering $\pi : V \to \{1, \ldots, n\}$, define the sequence distance $d^\pi_{\text{seq}}(u, v) := |\pi(u) - \pi(v)|$. The two-sided distortion can be written

$$D_{\text{seq}}(\pi) \;:=\; \frac{\displaystyle\max_{u \neq v} \frac{d^\pi_{\text{seq}}(u, v)}{d_G(u, v)}}{\displaystyle\min_{u \neq v} \frac{d^\pi_{\text{seq}}(u, v)}{d_G(u, v)}}.$$

Define $\rho_\pi(u, v) := d^\pi_{\text{seq}}(u, v)/d_G(u, v)$ for $u \neq v$, so that $D_{\text{seq}}(\pi) = \frac{\max \rho_\pi}{\min \rho_\pi}$. For any edge $\{u, v\} \in E$, $d_G(u, v) = 1$, hence $\rho_\pi(u, v) = |\pi(u) - \pi(v)|$. Therefore,

$$\max_{u \neq v} \rho_\pi(u, v) \;\geq\; \max_{\{u,v\} \in E} |\pi(u) - \pi(v)| \;=\; \varphi(\pi).$$

Let $x, y$ be the two adjacent vertices in $\pi$; then $d^\pi_{\text{seq}}(x, y) = 1$ while $d_G(x, y) \geq 1$, so

$$\min_{u \neq v} \rho_\pi(u, v) \;\leq\; \rho_\pi(x, y) = \frac{1}{d_G(x, y)} \;\leq\; 1.$$

Combining the two bounds proves the claim.

$$D_{\text{seq}}(\pi) = \frac{\max \rho_\pi}{\min \rho_\pi} \;\geq\; \frac{\varphi(\pi)}{1} \;\geq\; \varphi(G),$$

$\square$

### A.2    EXPRESSIVE LIMITATIONS OF SEQUENCE CANONICALIZATION

**Proposition A.2** ($\pi_V$ and $f_{\text{CanSeq}}$ are equally expressive). *Let $f_{\text{CanSeq}}$ be a canonical sequence–based model with universal $f_{\text{seq}}$ and let $\pi_V$ be its labeling function. Then, $f_{\text{CanSeq}} \simeq \pi_V$.*

*Proof.* Let $G_i = (V_i, E_i, X_i)$ for $i \in \{1, 2\}$ with $G_1 \not\cong G_2$. Let $\pi_V : V_i \to \mathbb{R}$ be a node labeler and define the *augmented* features $\tilde{\mathbf{x}}_v := (\mathbf{x}_v, \pi_V(v))$. Assume the augmented multisets coincide:

$$\{\{\tilde{\mathbf{x}}_v : v \in V_1\}\} \;=\; \{\{\tilde{\mathbf{x}}_v : v \in V_2\}\}.$$

Consider a single-sequence canonicalizer $\mathcal{C}_{\text{seq}}$ that outputs a permutation of $V$ and the corresponding sequence of per-node feature vectors, without adding structural annotations and whose ordering rule is a deterministic function of $\{\tilde{\mathbf{x}}_v\}_{v \in V}$ (e.g., a stable sort by a fixed key in $\tilde{\mathbf{x}}_v$ with deterministic tie-breaking depending only on $\tilde{\mathbf{x}}_v$). Because the two graphs have the same multiset of keys, and the ordering depends solely on these keys, the resulting ordered lists of features are identical:

$$\mathcal{C}_{\text{seq}}(G_1, \pi_V) \;=\; \mathcal{C}_{\text{seq}}(G_2, \pi_V).$$

(If ties occur, the tie-breaking is the same function of $\tilde{\mathbf{x}}$; when two items share identical $\tilde{\mathbf{x}}$, they are indistinguishable in the output sequence, so any permutation within such ties yields the same feature sequence.) Hence, there exist non-isomorphic graphs that collide under such $\mathcal{C}_{\text{seq}}$ that no $f_{\text{seq}}$ can distinguish, regardless of its expressivity. $\square$

**Example (DGCNN).** Let $\mathcal{C}_{\text{seq}} = \text{Sort}$ be a stable sort that orders vertices by a fixed key computed from $\tilde{\mathbf{x}}_v = (\mathbf{x}_v, \pi_V(v))$ with deterministic tie-breaking depending only on $\tilde{\mathbf{x}}_v$. If $\{\{(\mathbf{x}_v, \pi_V(v)) : v \in V_1\}\} = \{\{(\mathbf{x}_v, \pi_V(v)) : v \in V_2\}\}$ and $G_1 \not\cong G_2$, then $\text{Sort}(G_1, \pi_V) = \text{Sort}(G_2, \pi_V)$. This covers the DGCNN setting where $\pi_V \simeq f_{\text{MPNN}}$ provides the sort keys; the sort-based canonicalization cannot separate $G_1$ and $G_2$ beyond what is already encoded in the augmented multiset.

A.3 INVARIANT CANONICAL TREE NEURAL NETWORKS

We introduce notions of probabilistic invariance for random trees and covers, following definitions of probabilistic invariance for random walks (Kim et al., 2025) and searches (Ito et al., 2025).

**Definition A.3** (Probabilistic invariance for random trees). Let $\mathcal{A}$ be a randomized procedure that, on input a graph $G$, outputs a (labeled) spanning tree $T_{\mathcal{A}}(G)$. We say $\mathcal{A}$ is *probabilistically invariant* if for every pair of isomorphic graphs $G \cong_\pi H$ with isomorphism $\pi : V(G) \to V(H)$,

$$\pi\big(T_{\mathcal{A}}(G)\big) \stackrel{d}{=} T_{\mathcal{A}}(H).$$

**Definition A.4** (Probabilistic invariance for tree covers). Let $\mathcal{A}$ output a (multi)set or sequence of trees $\mathcal{T}_{\mathcal{A}}(G) = (T^{(0)}, \ldots, T^{(K-1)})$ on $G$. We call $\mathcal{A}$ *probabilistically invariant* if for every isomorphism $G \cong_\pi H$,

$$\pi\big(\mathcal{T}_{\mathcal{A}}(G)\big) \stackrel{d}{=} \mathcal{T}_{\mathcal{A}}(H),$$

where $\pi$ acts elementwise on the sequence (and, for an unordered cover, equality in distribution is taken after forgetting order).

**Lemma A.5** (MST is probabilistically invariant). *Let $G = (V, E)$ be an undirected graph. Let $w : E \to \mathbb{R}$ be an isomorphism–invariant base weight (so $w(g \cdot e) = w(e)$ for all $g \in \mathbb{S}_{|V|}$), and let $\zeta : E \to (0, 1)$ assign i.i.d. continuous tie–breakers to edges. Run Kruskal's algorithm (Algorithm 2) with lexicographic keys $k(e) = (w(e), \zeta(e))$ and let $X_{\mathrm{MST}}(G) = (e_0, \ldots, e_{|V|-2})$ be the resulting edge sequence. Then, for every $g \in \mathbb{S}_{|V|}$, $g \cdot X_{\mathrm{MST}}(G) \stackrel{d}{=} X_{\mathrm{MST}}(g \cdot G)$*

*Proof.* We prove by induction on $t$ that the $t$-th edge in Kruskal's sequence has the same *pushforward* conditional law on $G$ and on $g \cdot G$. For a prefix $\mathbf{x} = (e_0, \ldots, e_{t-1})$ valid for Kruskal on $G$, let $\mathcal{C}(G; \mathbf{x})$ be the component partition (union–find state) after processing $\mathbf{x}$. Define the *admissible set*

$$A(G; \mathbf{x}) := \{\, e = \{u, v\} \in E : \ u, v \text{ lie in different components of } \mathcal{C}(G; \mathbf{x}) \,\},$$

and the *frontier* of minimum–base–weight admissible edges

$$F(G; \mathbf{x}) := \{\, e \in A(G; \mathbf{x}) : \ w(e) = \min_{e' \in A(G; \mathbf{x})} w(e') \,\}.$$

Under Kruskal with keys $(w, \zeta)$, the next edge $e_t$ is the unique minimizer of $\zeta$ over $F(G; \mathbf{x})$. Since the $\zeta$'s are i.i.d. continuous, conditional on $\mathbf{x}$ the edge $e_t$ is *uniform* on $F(G; \mathbf{x})$.

**Base case** ($t = 0$). Here $A(G; \emptyset) = E$ and $F(G; \emptyset) = \{e \in E : \ w(e) = \min_{e' \in E} w(e')\}$. Because $w$ is isomorphism–invariant, $F(g \cdot G; \emptyset) = g \cdot F(G; \emptyset)$. The next edge is uniform on the respective frontier; pushing this uniform forward by $g$ yields

$$g \cdot X_{\mathrm{MST}}(G)[0] \stackrel{d}{=} X_{\mathrm{MST}}(g \cdot G)[0].$$

**Induction step.** Assume for some $t \geq 0$ that the prefixes satisfy

$$g \cdot X_{\mathrm{MST}}(G)[: t] \stackrel{d}{=} X_{\mathrm{MST}}(g \cdot G)[: t].$$

Fix any realization $\mathbf{x} = (e_0, \ldots, e_{t-1})$ of this prefix on $G$, and let $g\mathbf{x} = (g \cdot e_0, \ldots, g \cdot e_{t-1})$ be the corresponding prefix on $g \cdot G$. Relabeling preserves adjacency, hence components map as $\mathcal{C}(g \cdot G; g\mathbf{x}) = g \cdot \mathcal{C}(G; \mathbf{x})$ and therefore

$$A(g \cdot G; g\mathbf{x}) = g \cdot A(G; \mathbf{x}), \qquad F(g \cdot G; g\mathbf{x}) = g \cdot F(G; \mathbf{x}).$$

Conditional on $\mathbf{x}$, the next edge on $G$ is uniform on $F(G; \mathbf{x})$. Pushing this distribution forward by $g$ gives a uniform distribution on $g \cdot F(G; \mathbf{x}) = F(g \cdot G; g\mathbf{x})$, which is exactly the conditional law of the next edge on $g \cdot G$ given $g\mathbf{x}$:

$$g \cdot \big(X_{\mathrm{MST}}(G)[t] \,\big|\, \mathbf{x}\big) \stackrel{d}{=} X_{\mathrm{MST}}(g \cdot G)[t] \,\big|\, g\mathbf{x}.$$

Averaging over all realizations $\mathbf{x}$ (which, by the induction hypothesis, have matching laws under $G$ and $g \cdot G$ after applying $g$ to the $G$ prefix) yields

$$g \cdot X_{\mathrm{MST}}(G)[t] \stackrel{d}{=} X_{\mathrm{MST}}(g \cdot G)[t].$$

By induction for $t = 0, 1, \ldots, |V| - 2$, we conclude $g \cdot X_{\mathrm{MST}}(G) \stackrel{d}{=} X_{\mathrm{MST}}(g \cdot G)$, equivalently $X_{\mathrm{MST}}(G) \stackrel{d}{=} g^{-1} \cdot X_{\mathrm{MST}}(g \cdot G)$. $\square$

**Theorem A.6** (Probabilistic invariance of BUILDCANONICALTREECOVER). *Fix any isomor-phism–invariant node labeler $\pi_V$ (i.e., $\pi_V(g \cdot u) = \pi_V(u)$ for all $g \in \mathbb{S}_{|V|}$) and define base edge weights $\pi_E^{(0)}(u, v) = -(\pi_V(u) + \pi_V(v))$. For $k \geq 0$ let the iterative weights update be*

$$\pi_E^{(k+1)}(e) = \pi_E^{(k)}(e) + \tau \mathbf{1}\{e \in T^{(k)}\}, \qquad \tau > 0,$$

*where $T^{(k)}$ is the tree returned by KRUSKALMST (Algorithm 2) on $(G, \pi_E^{(k)})$ with i.i.d. continuous exchangeable tie–breakers $\zeta : E \to (0, 1)$ reused across all rounds. Let $\mathcal{T}(G) = (T^{(0)}, \dots, T^{(K-1)})$ be the random sequence of trees produced by BUILDCANONICALTREECOVER (Algorithm 1). Then, for every permutation $g \in \mathbb{S}_{|V|}$, $g \cdot \mathcal{T}(G) \overset{d}{=} \mathcal{T}(g \cdot G)$.*

*Proof.* We prove by induction on $k$ that, together with the evolving weights, the next tree is distribu-tionally equivariant under relabeling. Write the round–$k$ weights as a function of the history

$$\Pi^{(k)}(G) := \pi_E^{(k)}(\cdot; T^{(0)}, \dots, T^{(k-1)}),$$

and let $g$ act on edge–indexed objects by $(g \cdot f)(e) = f(g^{-1} \cdot e)$. The induction claim is

$$g \cdot T^{(k)}(G) \overset{d}{=} T^{(k)}(g \cdot G) \quad \text{and} \quad g \cdot \Pi^{(k+1)}(G) \overset{d}{=} \Pi^{(k+1)}(g \cdot G). \qquad (\star_k)$$

**Base case** ($k = 0$). Since $\pi_V$ is isomorphism–invariant, so is $\pi_E^{(0)}$: $\pi_E^{(0)}(g \cdot e) = \pi_E^{(0)}(e)$. By Lemma A.5, we have $g \cdot T^{(0)}(G) \overset{d}{=} T^{(0)}(g \cdot G)$. The update $\pi_E^{(1)} = \pi_E^{(0)} + \tau \mathbf{1}\{\cdot \in T^{(0)}\}$ is isomorphism–equivariant, hence $g \cdot \Pi^{(1)}(G) \overset{d}{=} \Pi^{(1)}(g \cdot G)$. Thus $(\star_k)$ holds for $k = 0$.

**Induction step.** Assume $(\star_k)$ holds for all $s < k$. Couple the two runs on $G$ and $g \cdot G$ by reusing the same exchangeable tie–breakers $\zeta$. By the induction hypothesis, the joint law of the prefixes

$$\left( g \cdot T^{(0)}(G), \dots, g \cdot T^{(k-1)}(G), g \cdot \Pi^{(k)}(G) \right)$$

equals the joint law of

$$\left( T^{(0)}(g \cdot G), \dots, T^{(k-1)}(g \cdot G), \Pi^{(k)}(g \cdot G) \right).$$

Conditioned on these prefixes, the round–$k$ Kruskal call on each graph uses (isomor-phism–equivariant) weights and the same i.i.d. continuous tie–breakers, so Lemma A.5 applies *conditionally* and yields

$$g \cdot \left( T^{(k)}(G) \mid T^{(<k)}(G), \Pi^{(k)}(G) \right) \overset{d}{=} T^{(k)}(g \cdot G) \mid T^{(<k)}(g \cdot G), \Pi^{(k)}(g \cdot G).$$

Averaging over the (matched) prefixes gives $g \cdot T^{(k)}(G) \overset{d}{=} T^{(k)}(g \cdot G)$. Finally, the weight update $\Pi^{(k+1)} = \Pi^{(k)} + \tau \mathbf{1}\{\cdot \in T^{(k)}\}$ is isomorphism–equivariant, so $g \cdot \Pi^{(k+1)}(G) \overset{d}{=} \Pi^{(k+1)}(g \cdot G)$. Thus $(\star_k)$ holds for round $k$. By induction for $k = 0, 1, \dots, K - 1$ we conclude $g \cdot \mathcal{T}(G) \overset{d}{=} \mathcal{T}(g \cdot G)$. □

Having established $\mathcal{T}(G)$ is probabilistically invariant, it follows that $f_{\text{CTNN}}(G)$, a deterministic function on $\mathcal{T}(G)$, is also probabilistically invariant.

### A.4 EXPECTED DISTORTION BOUNDS FOR CTNNS

**Theorem A.7** (UST expected distortion). *Let $G$ be a graph, and let $T$ be a uniform random spanning tree of $G$. Denote by $H(u, v)$ the random walk hitting time from $u$ to $v$. Then,*

$$D_{\text{UST}} = \max_{u,v} \mathbb{E}\left[d_T(u, v)\right]/d_G(u, v), \quad \mathbb{E}\left[d_T(u, v)\right] \leq H(u, v) + H(v, u)/2.$$

*Proof.* Run Wilson's algorithm (Wilson, 1996) rooted at $v$. The unique $u$–$v$ path in the resulting uniform spanning tree has the same distribution as the loop-erasure of a simple random walk started at $u$ and stopped upon first hitting $v$, $\text{LE}(S[0, \tau_v])$. Therefore,

$$d_T(u, v) \overset{d}{=} |\text{LE}(S[0, \tau_v])|.$$

Since loop-erasure only removes steps, $|\text{LE}(S[0, \tau_v])| \leq \tau_v$. Taking expectations yields

$$\mathbb{E}[d_T(u, v)] \leq \mathbb{E}_u[\tau_v] = H(u, v).$$

By symmetry, rooting Wilson's algorithm at $u$ also gives

$$\mathbb{E}[d_T(u, v)] \leq H(v, u).$$

Hence

$$\mathbb{E}[d_T(u, v)] \leq \min\{H(u, v), H(v, u)\} \leq \frac{H(u, v) + H(v, u)}{2} = \frac{C_{uv}}{2}.$$

$\square$

## A.5 COVERAGE AND EXPRESSIVITY GUARANTEES VIA MST CANONICALIZATION

**Lemma A.8** (Logarithmic spanning–tree cover). *Let $G = (V, E)$ be a graph with $m = |E|$ and arboricity $\Upsilon(G)$, the minimum number of forests required to cover $G$. Fix any node labeler $\pi_V$ with $\tau > \max_e \pi_E^{(0)}(e) - \min_e \pi_E^{(0)}(e)$. Denote $\mathcal{T} = \{T^{(k)}\}_{k=0}^{K-1}$ as the set of trees produced by a CTNN. If $K \geq \Upsilon(G) \ln m$ iterations, the union of the MSTs covers all edges: $\bigcup_{k=0}^{K-1} E(T^{(k)}) = E$.*

*Proof.* Let $\Upsilon = \Upsilon(G)$. By the definition of arboricity, there exists a partition of the edges into $\Upsilon$ forests, $E = \dot{\bigcup}_{j=1}^{\Upsilon} E(F_j)$. For each $j$, fix a (witness) spanning tree $\widetilde{T}_j \supseteq F_j$. Let $U_k \subseteq E$ be the set of *uncovered* edges after $k$ rounds and set $u_k := |U_k|$. For each $j$, define $u_{k,j} := |U_k \cap E(F_j)|$, so that $\sum_{j=1}^{\Upsilon} u_{k,j} = u_k$. By the pigeonhole principle, there exists $j^\star$ with $u_{k,j^\star} \geq u_k/\Upsilon$. Choose $\tau > \max_e b_e - \min_e b_e$. Then, for any $k \geq 1$,

$$\min_{e \in E \setminus U_k} \pi_E^{(k)}(e) \geq \max_{e \in U_k} \pi_E^{(k)}(e),$$

i.e., every seen edge is strictly more expensive than every unseen edge. Hence, minimizing total weight over spanning trees is equivalent to *maximizing* the number of unseen edges $|T \cap U_k|$ (any exchange that replaces a seen edge by an unseen edge strictly reduces cost). Since $\widetilde{T}_{j^\star}$ contains all edges of $F_{j^\star}$, it achieves $|\widetilde{T}_{j^\star} \cap U_k| = u_{k,j^\star}$. Therefore the MST $T^{(k)}$ satisfies

$$|T^{(k)} \cap U_k| \geq u_{k,j^\star} \geq \frac{u_k}{\Upsilon}.$$

Consequently,

$$u_{k+1} = u_k - |T^{(k)} \cap U_k| \leq \left(1 - \frac{1}{\Upsilon}\right) u_k.$$

Iterating yields $u_K \leq u_0 \left(1 - \frac{1}{\Upsilon}\right)^K \leq m \, e^{-K/\Upsilon}$. Choosing $K \geq \Upsilon \ln m$ gives $u_K < 1$, hence $U_K = \varnothing$ and $\bigcup_{k=0}^{K-1} E(T^{(k)}) = E$, as claimed. $\square$

**Proposition A.9** ($f_{\text{MPNN}} \prec f_{\text{CanTree}}$). *Let $f_{\text{CanTree}}$ be a CTNN satisfying Lemma 5.3, equipped with $\pi_V \simeq f_{\text{MPNN}}$. Then, $\forall G, H$, $f_{\text{CanTree}}(G) \overset{d}{=} f_{\text{CanTree}}(H) \implies f_{\text{MPNN}}(G) = f_{\text{MPNN}}(H)$. Moreover, $\exists G \not\cong H$ such that $f_{\text{CanTree}}(G) \overset{d}{\neq} f_{\text{CanTree}}(H)$ while $f_{\text{MPNN}}(G) = f_{\text{MPNN}}(H)$.*

*Proof.* We prove the preorder claim and then strictness.

**Step 1:** $f_{\text{MPNN}} \preceq f_{\text{CanTree}}$. We prove a stronger statement with deterministic distinguishability

$$f_{\text{CanTree}}(G) = f_{\text{CanTree}}(H) \implies f_{\text{MPNN}}(G) = f_{\text{MPNN}}(H).$$

Let $T^{(1)}(G)$ denote the first tree in the canonical cover produced on $G$ and similarly $T^{(1)}(H)$. Since the cover is aggregated with an *injective* multiset aggregator $f_{\text{agg}}$, equality $f_{\text{CanTree}}(G) = f_{\text{CanTree}}(H)$ implies equality of the aggregated elements, and in particular we may restrict attention to the contribution of the first tree.

Fix a graph $G = (V, E, \mathbf{X})$ and a spanning tree $T = T^{(1)}(G)$ with root chosen canonically. For $v \in V$, write $C_T(v)$ for its children, $p_T(v)$ for its parent (if $v$ is not the root), and $N_G(v)$ for its neighbors in $G$. A TreeMPNN update on the first tree can be written in the form

$$\mathbf{h}_v^{(T)} = f_{\text{agg}}\Big(\{\{\mathbf{x}_u : u \in C_T(v)\}\} \uplus \{\{\mathbf{x}_{p_T(v)}\}\} \uplus \{\{\mathbf{x}_u : u \in N_G(v) \setminus (C_T(v) \cup \{p_T(v)\})\}\}\Big),$$

where $f_{\text{agg}}$ is permutation-invariant and injective on multisets. By construction, the multiset union ranges over *exactly* the neighbors of $v$ in $G$ (partitioned into children, the parent, and residual neighbors), so no neighbor information is dropped. Importantly, however, the elements being aggregated are not necessarily the raw features $\mathbf{x}_u$: the first two groups correspond to *tree-contextual* representations. Concretely, child nodes contribute representations obtained by an injective map of their rooted subtrees *below* them, parent nodes contribute representations obtained by an injective map of the rooted tree *above* them, and the remaining non-tree neighbors contribute their original representations. Since compositions of injective functions are injective, the resulting TreeMPNN update at $v$ remains an injective function of the neighborhood multiset (now with context-dependent "colors"), and thus can simulate a standard injective message-passing update at $v$.

Now take any two graphs $G, H$ and suppose $f_{\text{CanTree}}(G) = f_{\text{CanTree}}(H)$ for a fixed realization of the CTNN randomness. Restricting to the first tree as above, for every node $v$ the corresponding per-node representations must coincide. Since the TreeMPNN update is a composition of injective functions on multisets, equality of the TreeMPNN outputs forces equality of the underlying neighborhood multisets, which in turn implies equality of the corresponding MPNN outputs. This proves the distributional implication $f_{\text{CanTree}}(G) \stackrel{d}{=} f_{\text{CanTree}}(H) \Rightarrow f_{\text{MPNN}}(G) = f_{\text{MPNN}}(H)$.

**Step 2: Strictness.** We exhibit connected graphs $G \not\cong H$ that are indistinguishable by MPNNs but separable by CTNNs. Let $G$ be the graph obtained by taking two $(n-1)$-cycles and connecting them by an additional single bridge edge, and let $H$ be obtained by taking two $n$-cycles and joining them (without introducing new edges) by a single bridge edge (Figure 6, $n = 6$). These graphs are stably colored identically by 1-WL, so $f_{\text{MPNN}}(G) = f_{\text{MPNN}}(H)$.

Initialize node labels and edge weights as in the CTNN construction, and consider the first MST extraction. By symmetry of the initialization, there is a unique valid MST in each graph obtained by deleting (from each cycle) the edge farthest from the bridge, yielding a spanning tree that preserves the bridge and breaks cycles at their most "distant" location. The resulting trees $T^{(1)}(G)$ and $T^{(1)}(H)$ are isomorphic. Consequently, a CTNN using a single canonical tree fails to distinguish $G$ and $H$.

Now assume $K$ satisfies Lemma 5.3, so the iterative reweighting scheme achieves full edge coverage across the cover. Consider the second round after reweighting edges missed at round 1. On $G$, there are exactly four possible MSTs at round 2, all isomorphic to each other (they are obtained by breaking, on each cycle, one of the two edges adjacent to the upweighted "missed" edge from round 1). On $H$, there are also four possible MSTs at round 2, but they fall into two non-isomorphic types (as illustrated in Figure 6). Therefore the distributions of the second tree differ:

$$T^{(2)}(G) \stackrel{d}{\neq} T^{(2)}(H).$$

Since the tree encoder and the multiset aggregator $f_{\text{agg}}$ are injective, a difference in distribution of one tree in the cover implies a difference in the distribution of the overall CTNN outputs, and thus

$$f_{\text{CanTree}}(G) \stackrel{d}{\neq} f_{\text{CanTree}}(H).$$

Combining with $f_{\text{MPNN}}(G) = f_{\text{MPNN}}(H)$ yields strictness: $f_{\text{MPNN}} \prec f_{\text{CanTree}}$. $\square$

To prove universality, we equip CTNNs with labels obtained from the canonical node labeler similar to the randomized anonymous labeling strategy used in (Wang and Cho, 2024; Kim et al., 2025). Intuitively, canonical labels allow separation among all graphs up to isomorphism.

**Definition A.10** (Canonical node labels). Let $G = (V, E, X)$ and let $\pi_V$ be a *canonical* node labeler, i.e., an isomorphism-invariant map $\pi_V : V \to \mathbb{R}$ such that $\pi_V$ assigns distinct labels to all nodes. Define the canonical ordering $\sigma_\pi : V \to \{1, \dots, |V|\}$ by sorting nodes by $\pi_V$: $\sigma_\pi^{-1}(i) = \text{argsort}\, \pi_V[i]$. We form *canonically labeled* node features $\tilde{\mathbf{x}}_v := (\mathbf{x}_v, \sigma_\pi(v))$, and use the same canonical labels for all trees in the CTNN cover. This yields a labeled cover

$$\mathcal{T}_\pi(G) := \{(T^{(k)}(G), \tilde{\mathbf{x}})\}_{k=0}^{K-1}.$$

**Lemma A.11** (Canonically labeled covers are separating). *Let $K$ satisfy Lemma 5.3 so that $\bigcup_k E(T^{(k)}(G)) = E(G)$. Assume $\pi_V$ separates all nodes on the finite class $\mathcal{G}$ (so $\sigma_\pi$ is well-defined). Then the map $G \mapsto \mathcal{T}_\pi(G)$ is separating up to isomorphism on $\mathcal{G}$.*

*Proof.* Let $\sigma_\pi$ be the canonical ordering induced by $\pi_V$. Form the canonical edge multiset

$$\mathrm{Canon}(G) := \{\!\{\, \{\, \sigma_\pi(u), \sigma_\pi(v) \,\} \; : \; \{u,v\} \in \bigcup_k E\big(T^{(k)}(G)\big) \,\}\!\}$$
$$= \{\!\{\, \{\, \sigma_\pi(u), \sigma_\pi(v) \,\} \; : \; \{u,v\} \in E(G) \,\}\!\}$$

where the equality uses Lemma 5.3. Since $\sigma_\pi$ is a bijection, $\mathrm{Canon}(G)$ is exactly the edge list of $G$ expressed in canonical node coordinates. Hence if $G \not\cong H$ then $\mathrm{Canon}(G) \neq \mathrm{Canon}(H)$, so the canonically labeled cover separates graphs in $\mathcal{G}$ up to isomorphism. □

**Theorem A.12** (CTNN Universality). *Let $\mathcal{G}$ be a finite class of graphs. Assume: (i) $K$ satisfies Lemma 5.3; (ii) the tree encoder $f_{\mathrm{tree}}$ and aggregation $f_{\mathrm{agg}}$ are universal on their domains; and (iii) $\pi_V$ is a canonical node labeler that separates all nodes on $\mathcal{G}$. Then for any continuous invariant graph function $f : \mathcal{G} \to \mathbb{R}$ and any $\varepsilon > 0$, there exists a CTNN such that*

$$\sup_{G \in \mathcal{G}} \big| f_{\mathrm{CTNN}}(G) - f(G) \big| \; \leq \; \varepsilon.$$

*Proof.* By Lemma 5.3, the cover edges union to $E(G)$; by Lemma A.11, the canonically labeled cover $\mathcal{T}_\pi(G)$ is separating on $\mathcal{G}$ via $\mathrm{Canon}(G)$. Thus the invariant target $f$ factors as

$$f(G) \; = \; F\big(\, \{\!\{\, \rho(T) : T \in \mathcal{T}_\pi(G) \,\}\!\} \,\big)$$

for some continuous permutation-invariant $F$ on multisets of tree representations and some $\rho$ on canonically labeled trees (e.g., any encoding that reproduces $\mathrm{Canon}(G)$). By universality, $f_{\mathrm{tree}}$ can approximate $\rho$ arbitrarily well on the finite support of observed canonically labeled trees, and $f_{\mathrm{agg}}$ can approximate $F$ on finite multisets of tree representations. Because $\mathcal{G}$ is finite, the composition error is uniformly bounded by $\varepsilon$ after suitable parameter choices. □

## B  ADDITIONAL MODEL DETAILS

### B.1  ALGORITHMS FOR CANONICAL SPANNING-TREE COVERS

Algorithm 1 outlines our procedure for building a $K$–tree canonical cover. We initialize the edge labeler $\pi_E^{(0)}$ using the negative sum of endpoint degrees for each edge, which prioritizes edges incident to high-degree nodes. For rounds $t = 0, \ldots, K - 1$, we construct an MST with respect to the current edge weights using Kruskal's algorithm (Algorithm 2): edges are stably sorted (random tie–breaking) and scanned in nondecreasing order, adding an edge if it does not create a cycle. After forming the tree $T^{(t)}$, we update the labeler to encourage coverage in subsequent rounds: edges *not* selected in $T^{(t)}$ receive an additive penalty (controlled by $\tau$) that increases their priority in the next MST. Finally, we choose a canonical root as the tree center via the standard two–BFS routine (Algorithm 3): one BFS finds an endpoint of a longest path, and a second BFS from that endpoint finds the opposite endpoint; the center(s) of this path serve as the root. This procedure runs in $O(m \log n)$ per round for the MST and $O(n)$ for root selection, and returns the $K$ trees with their canonical roots.

---

**Algorithm 1:** BUILDCANONICALTREECOVER: iterative MST cover with root selection

---

**Input:** Graph $G = (V, E)$; node labeler $\pi_V : V \to \mathbb{R}$; iterations $K$; step $\tau > 0$; tiny $\varepsilon > 0$
**Output:** Tree cover $\mathcal{T} = \{T^{(k)}\}_{k=0}^{K'-1}$ and roots $\mathcal{R} = \{r^{(k)}\}_{k=0}^{K'-1}$
**(Initialization)**
 **foreach** $e = \{u, v\} \in E$ **do**
  $w_e^{(0)} \leftarrow -\big(\pi_V(u) + \pi_V(v)\big)$     `/* base edge weights `$\pi_E^{(0)}$` */`

 Draw i.i.d. tie–breakers $\zeta : E \to (0, 1)$ (fixed across rounds)
 $S_0 \leftarrow \varnothing$         `/* covered edges so far */`
 $\mathcal{T} \leftarrow \varnothing, \mathcal{R} \leftarrow \varnothing$
**for** $k = 0$ **to** $K - 1$ **do**
 `// 1) Minimum spanning tree with lexicographic keys`
 $T^{(k)} \leftarrow \text{KRUSKALMST}\big(G, \ e \mapsto \big(w_e^{(k)}, \zeta(e)\big)\big)$
 $\mathcal{T} \leftarrow \mathcal{T} \cup \{T^{(k)}\}; \quad S_{k+1} \leftarrow S_k \cup E(T^{(k)})$
 `// 2) Canonical root:  tree center via two BFS passes`
 $r^{(k)} \leftarrow \text{TREECENTER}\big(T^{(k)}, \pi_V\big)$
 $\mathcal{R} \leftarrow \mathcal{R} \cup \{r^{(k)}\}$
 `// 3) Edge-weight update (penalize edges just used)`
 **foreach** $e \in E$ **do**
  **if** $e \in E(T^{(k)})$ **then** $w_e^{(k+1)} \leftarrow w_e^{(k)} + \tau$
  **else** $w_e^{(k+1)} \leftarrow w_e^{(k)}$
 **if** $|S_{k+1}| = |E|$ **then break**

**return** $(\mathcal{T}, \mathcal{R})$

---

---

**Algorithm 2:** KRUSKALMST with exchangeable tie–breakers

---

**Input:** Undirected graph $G = (V, E)$; base edge weights $w : E \to \mathbb{R}$; tie–breakers $\zeta : E \to (0, 1)$ i.i.d.
**Output:** A spanning tree $T$ of $G$
$T \leftarrow \varnothing$; initialize disjoint–set $D$ with MAKESET$(v)$ for all $v \in V$
**if** $E = \varnothing$ **then return** $T$

`// I.i.d. continuous tie-breakers `$\zeta$` m ake keys distinct w.p. 1.`
**Sort** edges $E$ by nondecreasing key $k(e) := \big(w(e), \zeta(e)\big)$
`// Union-find tracks connected components of the partial forest.`
**for** $e = \{u, v\}$ *in $E$ in the above order* **do**
 **if** FIND$_D(u) \neq$ FIND$_D(v)$ **then**
  $T \leftarrow T \cup \{e\}; \quad$ UNION$_D(u, v)$
  **if** $|T| = |V| - 1$ **then return** $T$

**return** $T$

---

---

**Algorithm 3:** TREECENTER: root selection by two BFS passes

---

**Input:** Tree $T = (V_T, E_T)$; tie–breaker ranking on vertices (e.g., $(\pi_V, \text{ID})$)
**Output:** Root $r \in V_T$ (a center of $T$)
Pick canonical start $s \in V_T$ (minimizing the tie–breaker);
$u \leftarrow \text{BFS\_FARTHEST}(T, s);\quad v \leftarrow \text{BFS\_FARTHEST}(T, u);$
$P \leftarrow$ unique path from $u$ to $v$ in $T$;
**if** $|P|$ *odd* **then** $r \leftarrow$ middle vertex of $P$
**else** $r \leftarrow$ the nearer of the two middle vertices under the tie–breaker
**return** $r$

---

### B.2 ALGORITHMS FOR RECURRENT TREE NEURAL NETWORKS

We provide discussion and implementation details of the recurrent tree neural network (Algorithm 4).

---

**Algorithm 4:** BITREELSTMFORWARD: Bidirectional child–sum Tree-LSTM forward pass

---

**Input** : $x \in \mathbb{R}^{N \times D}$ node features; rooted tree $T = (V, E_T, r)$ in COO form with
$\quad\quad$ $(\text{row}, \text{col}) = (\text{parent}, \text{child})$; arrays $\text{parent}[v]$, $\text{depth}[v] \in \{0, \dots, L\}$
**Output** : $h \in \mathbb{R}^{N \times 2H}$: concat. of bottom-up and top-down hidden states

**Parameters:** bottom-up $W_{iou} \in \mathbb{R}^{D \times 3H}, U_{iou} \in \mathbb{R}^{H \times 3H}, W_f \in \mathbb{R}^{D \times H}, U_f \in \mathbb{R}^{H \times H}$; top-down
$W_{iou}^{\downarrow}, U_{iou}^{\downarrow}, W_f^{\downarrow}, U_f^{\downarrow}$ of matching shapes.

**Init:** $h^{\uparrow} \leftarrow 0_{N \times H}, c^{\uparrow} \leftarrow 0_{N \times H}, h^{\downarrow} \leftarrow 0_{N \times H}, c^{\downarrow} \leftarrow 0_{N \times H}.$
Bucket nodes by depth: $\mathcal{V}_\ell \leftarrow \{v \in V : \text{depth}[v] = \ell\}$ for $\ell = 0, \dots, L.$

*/\* Bottom-up pass (children $\rightarrow$ parent): process parents from deepest to root. \*/*
**for** $\ell = L$ **to** $0$ **do**
$\quad P \leftarrow \mathcal{V}_\ell$ $\quad\quad\quad\quad\quad\quad\quad\quad\quad\quad\quad\quad\quad$ `// parents at depth ℓ`
$\quad E_\ell \leftarrow \{(u \leftarrow v) \in E_T : u \in P\}$ $\quad\quad\quad\quad$ `// edges with parent at depth ℓ`
$\quad$ `// Aggregate child states with` **scatter_add** `(child-sum` $f_{\text{agg}} = \sum$`)`
$\quad h_{\text{sum}} \leftarrow 0_{N \times H};$

$\quad h_{\text{sum}}[u] \mathrel{+}= \sum_{(u \leftarrow v) \in E_\ell} h^{\uparrow}[v]$
$\quad$ `// Per-edge forgets and summed transformed cell contributions`
$\quad f_{uv} \leftarrow \sigma\big(W_f x[u] + U_f h^{\uparrow}[v]\big)$ for $(u \leftarrow v) \in E_\ell$
$\quad c_\sim \leftarrow 0_{N \times H};$

$\quad c_\sim[u] \mathrel{+}= \sum_{(u \leftarrow v) \in E_\ell} f_{uv} \odot c^{\uparrow}[v]$
$\quad$ `// Node-level gates and updates for all` $u \in P$
$\quad$ **for** $u \in P$ **do**
$\quad\quad [i, o, \tilde{u}] \leftarrow \text{split}_3\big(W_{iou} x[u] + U_{iou} h_{\text{sum}}[u]\big);$
$\quad\quad i \leftarrow \sigma(i),\ o \leftarrow \sigma(o),\ \tilde{u} \leftarrow \tanh(\tilde{u});$
$\quad\quad c^{\uparrow}[u] \leftarrow i \odot \tilde{u} + c_\sim[u];$
$\quad\quad h^{\uparrow}[u] \leftarrow o \odot \tanh\big(c^{\uparrow}[u]\big)$

*/\* Top-down pass (parent $\rightarrow$ children): propagate from root to leaves. \*/*
**for** $\ell = 1$ **to** $L$ **do**
$\quad V \leftarrow \mathcal{V}_\ell$ $\quad\quad\quad\quad\quad\quad\quad\quad\quad\quad\quad\quad\quad\quad$ `// children at depth ℓ`
$\quad$ **for** $v \in V$ **do**
$\quad\quad p \leftarrow \text{parent}[v]$ $\quad\quad\quad\quad\quad\quad\quad\quad\quad\quad$ `// unique parent`
$\quad\quad [i, o, \tilde{u}] \leftarrow \text{split}_3\big(W_{iou}^{\downarrow} x[v] + U_{iou}^{\downarrow} h^{\downarrow}[p]\big);$
$\quad\quad i \leftarrow \sigma(i),\ o \leftarrow \sigma(o),\ \tilde{u} \leftarrow \tanh(\tilde{u});$
$\quad\quad f \leftarrow \sigma\big(W_f^{\downarrow} x[v] + U_f^{\downarrow} h^{\downarrow}[p]\big);$
$\quad\quad c^{\downarrow}[v] \leftarrow i \odot \tilde{u} + f \odot c^{\downarrow}[p];$
$\quad\quad h^{\downarrow}[v] \leftarrow o \odot \tanh\big(c^{\downarrow}[v]\big)$

**return** $h \leftarrow \text{concat}\big(h^{\uparrow}, h^{\downarrow}\big)$ $\quad\quad\quad\quad\quad\quad\quad\quad\quad$ `// [N, 2H]`

---

**Parallel bottom–up / top–down passes.** For each canonical tree we precompute three arrays: `edge_index_tree` (row=parent, col=child), `parent[v]` (unique parent), and `depth[v]` (distance to root). The bidirectional forward pass runs in two levelwise sweeps that are parallel across all nodes at the same depth. In the bottom–up pass (leaves $\rightarrow$ root), we bucket nodes by `depth` and use `scatter_add` to implement the child–sum aggregator and edgewise forget contributions in a single batched operation over all edges whose parent is at the current depth. In the top–down pass (root $\rightarrow$ leaves), each node reads its parent's state via `index_select` and applies the same batched gating. This organization avoids Python loops over edges and exploits segmented reductions on the GPU; it only iterates over depth buckets. The forward pass can be efficiently batched across trees by treating the full batch as a collection of disjoint graphs, whose edges are stored in COO format.

**Tree encoder.** We implement a *bidirectional child–sum Tree–LSTM* layer with two parameter sets (children$\rightarrow$parent and parent$\rightarrow$children). Each direction computes input/output/update gates via a single linear projection that yields $3H$ channels per node and applies elementwise nonlinearity; edgewise forget gates are computed in parallel across all incident edges at a depth. The output feature per node is the concatenation of the two hidden states, making the layer stackable (we use residual/normalization as in standard practice when beneficial).

**Complexity and memory.** Each direction runs in $O(|V|)$ time and $O(|V|)$ memory for node states. For a cover of $K$ trees, the bidirectional layer cost is $O(K|V|)$. In practice we batch trees along the sample dimension, so the work parallelizes across graphs and across trees.

## C    DESIGN SPACE OF CANONICAL APPROACHES

As shown in Table 4, we organize canonicalization methods along six axes: (i) whether they rely on domain knowledge; (ii) the node labeler $\pi_V$; (iii) the edge labeler $\pi_E$; (iv) the canonicalizer (ordering/selection rule); (v) the induced representation (vector/sequence/tree); and (vi) whether they use a set of canonical elements per graph (vs. a single representative), together with the downstream encoder. Table 4 situates common pipelines: domain-driven approaches (Fingerprint, SMILES, Primary Seq.) produce a single canonical vector or sequence; graph-agnostic orderings (DGCNN/SortPooling, RCM) also yield a single sequence from graph-derived ranks. CTNN is the only approach that (a) uses an edge labeler to drive a coverage-aware canonicalization and (b) represents each graph by a tree cover, a set of canonical trees obtaining full coverage. This design is domain-agnostic, preserves distances more faithfully than sequences, and increases expressivity by operating on a set rather than a single representative.

Table 4: Design space for graph canonicalization. "Set" indicates whether the method uses multiple canonical elements per graph (e.g., a cover of trees) rather than a single canonicalization. "Domain Knowledge" indicates reliance on domain-specific information (e.g., chemistry rules).

| Approach | Domain Knowledge | Node Labeler | Edge Labeler | Canonicalizer | Representation | Set | Backbone |
|---|---|---|---|---|---|---|---|
| **Fingerprint** | Yes | NA | NA | Handcrafted chemical descriptors | Vector | No | MLP |
| **SMILES** | Yes | Atom canonical ranks | NA | Canonical SMILES algorithm | Sequence | No | RNN/TRSF |
| **Primary Seq.** | Yes | NA | NA | Identity | Sequence | No | RNN |
| **DGCNN** | No | MPNN | No | Differentiable sort (SortPooling) | Sequence | No | 1D CNN |
| **RCM** | No | Degree | No | Reverse Cuthill–McKee ordering | Sequence | No | RNN |
| **CTNN (full)** | No | Degree | Coverage-aware | Minimum Spanning Tree | Tree Cover | Yes | TreeMPNN |

## D    ADDITIONAL EXPERIMENTAL DETAILS

**Training and Hyperparameter Selection.**    All models are trained by minimizing the binary cross-entropy loss on binary classification tasks and the negative log-likelihood loss on multiclass classification tasks. Training is performed for a maximum of 200 epochs with early stopping patience set to 15 epochs based on validation performance. The best-performing model on the validation set is selected for evaluation on the test set. We perform a grid search over the following hyperparameters for models where applicable:

- Number of layers: $\{1, 2, 3, 4\}$
- Learning rate: $\{0.05, 0.01, 0.005, 0.001\}$
- Batch size: $\{64, 128, 512, 1024\}$
- Hidden dimension: $\{64, 128, 256\}$
- Global pooling: $\{\texttt{mean}, \texttt{sum}, \texttt{max}\}$
- Sequence model: $\{\texttt{GRU}, \texttt{LSTM}, \texttt{Transformer}\}$
- Number of sequences/trees $K$: $\{1, 4, 8\}$
- Coverage penalty $\tau$: $\{1, 2, 4\}$

All models are optimized using the Adam optimizer.

## E    EXTENDED RESULTS

We include additional analyses on (i) sensitivity to the node labeler $\pi_V$, (ii) the coverage penalty $\tau$, and (iii) the number of trees $K$. We find CTNN to be robust to the choice of $\pi_V$ and $\tau$, while increasing $K$ consistently increases coverage, reduces distortion, and improves performance. We further conduct a sensitivity analyses to the choice of sequence models for SMILES, comparing performance when $f_{\text{seq}}$ is a LSTM or transformer. Here, we find recurrence outperforms attention aligning with recent findings in RWNN studies. We lastly report preprocessing runtimes, confirming that CTNN's preprocessing is efficient and negligible with respect to training times for all datasets.

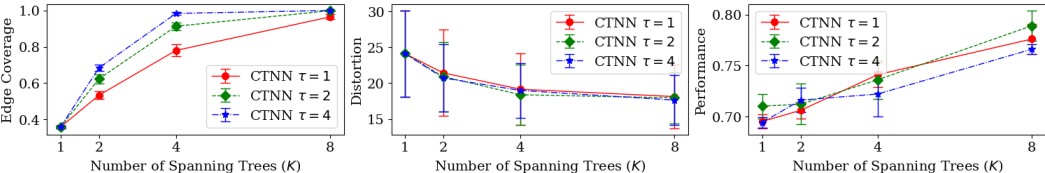

Figure 7: Sensitivity of CTNN to the number of trees $K$ and coverage penalty $\tau$ on GO BIO. Coverage rises rapidly with $K$; distortion decreases monotonically with $K$; and performance improves. Trends are similar across $\tau$, indicating robustness, with larger $\tau$ yielding slightly higher coverage at fixed $K$ on proteins. Error bars denote standard deviation over samples.

Table 5: Median (min, max) test AUC on molecular datasets. We test CTNN with three node labelers $\pi_V$: Degree, Closeness Centrality, BLISS. CTNN is robust across all choices of node labeler $\pi_V$.

| | | Small MoleculeNet Molecular Benchmarks (AUC ↑) | | | | | |
|---|---|---|---|---|---|---|---|
| | | **ClinTox** | **SIDER** | **BACE** | **BBBP** | **Tox21** | **ToxCast** |
| **# Graphs** | | 1.5K | 1.5K | 1.5K | 2K | 7.8K | 8.5K |
| **Avg. $\lvert V\rvert$** | | 26.1 | 33.6 | 34.1 | 23.9 | 16.4 | 17.1 |
| **Avg. $\lvert E\rvert$** | **Node Labeler $\pi_V$** | 28.0 | 35.4 | 36.9 | 26.0 | 16.9 | 17.5 |
| **CTNN** | Degree | 84.7 (78.5, 91.0) | 65.8 (62.7, 70.4) | 79.3 (75.4, 85.0) | 86.1 (80.6, 90.4) | 73.4 (69.6, 77.2) | 80.0 (75.4, 80.3) |
| **CTNN** | Centrality | 85.5 (78.0, 88.2) | 65.1 (62.6, 70.2) | 79.5 (73.1, 84.4) | 85.6 (77.1, 88.4) | 73.0 (72.8, 74.7) | 79.3 (76.5, 80.6) |
| **CTNN** | BLISS | 84.7 (78.5, 91.0) | 65.2 (61.5, 68.5) | 79.4 (71.8, 84.4) | 86.7 (80.5, 90.7) | 73.9 (69.4, 79.2) | 79.5 (76.7, 80.7) |

## E.1 SENSITIVITY ANALYSES

**Sensitivity to $K$.** We vary the number of trees $K$ on GO BIO (Figure 7). Edge coverage increases rapidly with $K$, reaching full coverage by $K{=}8$ on GO BIO, consistent with the theory that only a small number of trees is needed on sparse graphs. Distortion decreases monotonically as $K$ grows, indicating that additional trees better preserve original graph distances. Task performance likewise improves with $K$, showing the practical value of the canonical tree cover.

**Sensitivity to $\tau$.** We test coverage penalty $\tau = \{1, 2, 4\}$ on the same dataset (Figure 7). Across benchmarks, coverage, distortion, and accuracy follow similar trends for different $\tau$, indicating robustness to the choice of penalizer. For proteins, larger $\tau$ yields higher coverage at a fixed $K$, as heavier penalties bias the MST toward previously unseen edges. Overall, CTNN's behavior is stable across $\tau$, while $K$ primarily controls the coverage–distortion–accuracy tradeoff.

**Sensitivity to $\pi_V$.** We evaluate CTNN with three choices of node labeler $\pi_V$ on small-molecule benchmarks: Degree, Closeness Centrality, and BLISS (Table 5). Across all datasets, performance is consistent across labelers, indicating that CTNN is *robust* to the choice of $\pi_V$ in this regime. This suggests that, once the canonical tree cover is constructed and processed with an expressive tree encoder, CTNN can recover strong performance even with inexpensive labelers. Accordingly, we default to Degree in our main experiments for efficiency, while Centrality or BLISS provide comparable results at higher preprocessing cost.

**Sensitivity to $f_{\text{seq}}$.** We evaluate the sensitivity of the sequence encoder $f_{\text{seq}}$ by comparing attention (Transformer) and recurrence (LSTM) on canonical SMILES, and include a graph Transformer (GT) that operates directly on molecular graphs (Table 6). We report the analysis on the molecular benchmarks, whereas training analogous models on the larger, denser protein graphs did not converge within 24 hours. Results demonstrate that attention and recurrence perform comparably on SMILES and are similar to the GT baseline. In all cases, however, these models underperform in comparison to CTNNs, which maintain a clear performance advantage. This occurs since attention relies on the sequential positional encoding and recurrence relies on the linear ordering, which both incur distortion and fail to capture graph distances

Table 6: Transformers and LSTMs achieve comparable AUC across PCBA datasets, indicating that attention and recurrence perform similarly on the canonical sequence. GTs also perform comparably to both. Values are median (min, max) over splits. CTNNs outperform all models.

| | | Molecular Benchmarks (AUC $\uparrow$) | | | |
|---|---|---|---|---|---|
| Approach | Backbone | PCBA-1030 | PCBA-1458 | PCBA-4467 | PCBA-5297 |
| GT | Transformer | 68.1 (67.9, 68.6) | 81.2 (81.0, 81.5) | 78.9 (77.8, 79.9) | 87.7 (87.6, 88.2) |
| SMILES | Transformer | 71.9 (71.5, 72.3) | 84.4 (83.7, 84.5) | 82.4 (81.5, 82.5) | 88.9 (88.3, 89.4) |
| SMILES | LSTM | 71.9 (71.2, 72.5) | 84.9 (84.5, 85.9) | 81.1 (80.0, 81.4) | 90.2 (90.0, 90.3) |
| CTNN | TreeMPNN | **80.6 (80.3, 81.2)** | **89.1 (88.0, 89.9)** | **86.8 (86.5, 87.4)** | **94.6 (94.2, 94.9)** |

## E.2 RUNTIME ANALYSES

We report per-graph preprocessing time for CTNNs when constructing $K=8$ trees with a degree-based node labeler $\pi_V(v) = \deg(v)$ (Table 7). On molecular graphs the cost is in the milliseconds, and on protein graphs it is on the order of tenths of a second. In practice, this preprocessing is parallelizable across graphs and is computed once and reused over all training epochs, making it a small fraction of end-to-end training time. Overall, CTNN preprocessing is efficient for the datasets considered.

Table 7: CTNN preprocessing time per graph to construct $K=8$ canonical spanning trees using degree labeler ($\pi_V(v) = \deg(v)$). We report dataset sizes and average graph statistics; times (seconds) are averaged over all graphs. Overall, CTNN preprocessing is efficient across all datasets.

| Dataset | # Graphs | Avg. $|V|$ | Avg. $|E|$ | Avg. $\deg(v)$ | Time (sec) |
|---|---|---|---|---|---|
| PCBA-1030 | 160K | 24.3 | 26.2 | 2.2 | 0.004 |
| GO MOL | 32K | 250.1 | 687.5 | 5.4 | 0.093 |
| NeuroGraph | 7K | 400.0 | 7000.29 | 17.6 | 0.127 |

## E.3 UST VS. CTNN DISTORTION

The distortion bound in Theorem A.7 is stated for uniform spanning trees (USTs), whereas CTNN constructs coverage-aware minimum spanning trees (MSTs) from a different, non-uniform distribution. USTs are used in the theory as a well-understood reference distribution to motivate why random tree covers can achieve low distortion. Empirically, however, we find that our coverage-aware MST scheme achieves even lower distortion than USTs on the datasets we consider. Table 8 compares the shortest-path distortion for USTs and CTNN (degree labeler, $\tau=1$) on PCBA-1030 as the number of trees $K$ increases. CTNN consistently attains smaller average distortion and variance than USTs for all $K$. This advantage is plausibly due to the coverage-aware construction, which explicitly favors edges that have not appeared in earlier trees, leading to tree covers that better preserve graph distances than independent UST samples, which have no bias toward uncovered edges. These results provide quantitative evidence that the practical CTNN tree distribution inherits and can even improve upon the low-distortion behavior suggested by the UST analysis.

Table 8: Distortion as the number of trees $K$ increases on PCBA-1030 for uniform spanning trees (UST) and CTNN, using a degree-based node labeler and $\tau=1$. We report mean $\pm$ standard deviation distortion over graphs.

| | PCBA-1030 Distortion | | | |
|---|---|---|---|---|
| | $K=1$ | $K=2$ | $K=4$ | $K=8$ |
| UST | $5.58 \pm 1.23$ | $4.49 \pm 1.16$ | $3.57 \pm 0.70$ | $2.92 \pm 0.45$ |
| CTNN | $5.10 \pm 0.45$ | $3.08 \pm 0.30$ | $2.26 \pm 0.42$ | $2.13 \pm 0.26$ |

### E.4 Evaluation on Non-Molecular Benchmarks

We ran additional experiments on a non-molecular and non-protein benchmark (Table 9). Our main evaluation focuses on molecular and protein graph classification tasks because canonicalization is widely adopted and frequently used in practice in these domains. To demonstrate that our approach is applicable beyond these biochemical domains, we additionally evaluate CTNN on a brain graph classification benchmark from NeuroGraph (Said et al., 2023). Across this benchmark, CTNN obtains the best performance compared to a standard MPNN (GIN) and canonicalization baselines (RCM and DGCNN). These results demonstrate CTNNs' applicability to domains in which canonicalization is not yet widely adopted.

Table 9: Median (min, max) accuracy of GIN, RCM, DGCNN, and CTNN on the NeuroGraph Activity prediction benchmark. CTNN obtains the best performance across baselines.

|  | NeuroGraph-Activity |
|---|---|
| **# Graphs** | 7K |
| **Avg. $|V|$** | 400 |
| **Avg. $|E|$** | 7000 |
| **Avg. deg** | 18.0 |
| **Metric** | ACC ↑ |
| **GIN** | 85.4 (85.4, 86.1) |
| **RCM** | 91.5 (91.3, 92.5) |
| **DGCNN** | 91.9 (91.9, 92.5) |
| **CTNN** | **94.1 (94.0, 94.2)** |

