# OpenReview forum: "Canonical Tree Cover Neural Networks for Expressive and Invariant Graph Learning"
_ICLR.cc/2026/Conference — ICLR 2026 Poster_

### Official Review · Reviewer_vjKg · 2025-10-27

**Soundness:** 2
**Presentation:** 4
**Contribution:** 3
**Rating:** 4
**Confidence:** 5

**Summary:**

This paper proposes Canonical Tree Cover Neural Networks (CTNNs) as a new approach to graph canonicalization. Instead of flattening graphs into a single sequence (which causes distortion and expressivity loss), CTNNs construct a canonical spanning tree cover and process each tree with expressive tree encoders, aggregating their outputs into an invariant representation. The authors provide theoretical results (distance preservation, expressivity beyond 1-WL, universality under certain conditions) and empirical evaluation on molecular and protein benchmarks, showing improvements over message-passing GNNs, sampling approaches, and sequence canonicalization baselines.

**Strengths:**

1. The paper clearly explains the limitations of sequence canonicalization, illustrating them with intuitive examples such as star graphs.
2. It introduces a tree cover method that better preserves structural information and invariance, supported by formal analyses of distortion, expressivity, and coverage guarantees.
3. On multiple benchmarks, CTNNs consistently surpass strong baselines, achieving notable gains on molecular tasks and competitive results on protein tasks.

**Weaknesses:**

1. The key idea (use a set of trees instead of one sequence) feels like an incremental extension rather than a major conceptual advance. Many theoretical results (universality, expressivity boost by multiple views, distortion comparisons) are natural consequences of existing work. Frequent use of terms like “strictly more expressive,” “provably invariant,” and “universal” gives an impression of overselling. Some proofs (e.g., universality of CTNNs) rely on standard functional approximation arguments, which are not truly new.

2. The guarantees, such as using only a logarithmic number of trees and achieving low distortion, mainly hold for sparse graphs. Dense graphs, such as proteins, show weaker improvements and less favorable theoretical bounds. Although the paper claims MST construction is efficient, it does not provide runtime or memory benchmarks. For dense graphs, the (O(Km \log n)) cost could be significant, and the preprocessing advantage is asserted but not supported with quantitative evidence.

3. Improvements on molecular benchmarks are often only 1–2 AUC points. On protein tasks, performance is inconsistent, sometimes close to baselines. No large-scale or real-world datasets beyond the standard benchmarks are tested. The ablation studies show predictable drops (removing key components hurts), but provide little insight into why certain parts matter. No analysis of failure cases or adversarial graph structures.

4. While molecular and protein benchmarks are reasonable, the method is not shown on social networks, knowledge graphs, or large heterogeneous graphs. This raises questions about generality beyond biochemical datasets.

**Questions:**

I will raise my score if the authors address W2.

---

> ### Author Response · Authors · 2025-11-21
> **Responses to Reviewer vjKg Part 1**
>
> We thank the reviewer for the thoughtful and positive evaluation. We are glad that the limitations of sequence-based canonicalization and the motivating examples (e.g., star graphs) came across clearly, and that the tree-cover perspective of CTNNs is seen as a principled way to better preserve structure and invariance. We also appreciate the recognition of our formal analyses on distortion, expressivity, and coverage, as well as the empirical gains over strong baselines on molecular tasks and the competitive performance on protein benchmarks. Below, we address your specific comments individually.
>
> - **CTNN Novelty.** While prior work has shown that decomposing graphs into arbitrary subgraphs can boost expressivity, our distortion perspective is new: we formally show that any single-sequence canonicalization incurs distortion lower bounded by the graph bandwidth, and we derive complementary expected-distortion bounds for spanning tree distributions, explaining why a tree cover preserves graph geometry better than a single sequence. These distortion results are not direct consequences of existing work on subgraph GNNs, and they are what make CTNNs unique in comparison to existing subgraph GNNs and canonicalization approaches
>
> - **CTNN Runtime and Memory**. We report quantitative preprocessing times in Appendix E.2 (Table 6). There we measure the average CTNN preprocessing time per-graph to construct $K=8$ spanning trees with a degree-based node labeler. On molecular graphs the cost is in the milliseconds, and on protein graphs it is on the order of tenths of a second. To demonstrate our approach is scalable to larger graphs, we have since included a runtime analysis on brain graphs from NeuroGraph [1], which have substantially more edges than our molecule and protein graphs. On these graphs CTNN preprocessing completes in $\approx 0.13$ seconds per graph on average. In practice, this preprocessing is fully parallelizable across graphs and is performed once and reused for all training epochs, so it contributes only a small fraction of end-to-end training time. These results demonstrate CTNN’s scalability and efficiency across a spectrum of graphs, including the larger NeuroGraph setting. Because we fix a small constant $K=8$, storing the tree cover costs only $O(Kn)$ edges per graph, making the memory overhead modest for all datasets considered.
>
> **Table 6.** CTNN preprocessing time per graph to construct $K = 8$ canonical spanning trees using the degree labeler $\pi_V(v) = \deg(v)$. We report dataset sizes, average graph statistics, and average preprocessing time (seconds) per graph. CTNN preprocessing is efficient across all datasets.
>
> | Dataset        | # Graphs | Avg. Nodes | Avg. Edges | Avg. deg(v) | Time (sec) |
> |---------------|----------|------------|------------|-------------|------------|
> | **PCBA-1030** | 160K     | 24.3       | 26.2       | 2.2         | 0.004      |
> | **GO MOL**    | 32K      | 250.1      | 687.5      | 5.4         | 0.093      |
> | **NeuroGraph**| 7K       | 400.0      | 7000.29    | 17.6        | 0.127      |
>
>
> - **Evaluation on non-molecule and non-protein benchmarks.** We ran additional experiments on non-molecular and non-protein benchmarks (Table 7). We primarily focus our evaluation on molecular and protein graph classification tasks because canonicalization is widely adopted and frequently used in practice in these domains. In contrast, standard MPNNs already perform well on single-graph node-classification tasks on social networks, where canonicalization is likely to offer limited benefit. To demonstrate that our approach is applicable beyond biochemical datasets, we additionally evaluate CTNN on a brain graph classification benchmark from NeuroGraph [1]: an Activity prediction set of 7K brain graphs (avg. 400 nodes, 7K edges, avg. degree ≈ 18) where the task is to predict one of seven mental states (e.g., emotion processing, language). Across the benchmark, CTNNs obtain the best performance compared to a standard MPNN and canonicalization baselines. These results demonstrate CTNNs’ applicability to domains in which canonicalization is not yet widely adopted. We have added these results on NeuroGraph to the paper.
>
> **Table 7.** Median (min, max) accuracy of GIN, RCM, DGCNN, and CTNN on the NeuroGraph Activity benchmark.
>
> | Method | Activity (ACC)     |
> |--------|--------------------|
> | GIN    | 85.4 (85.4, 86.1)  |
> | RCM    | 91.5 (91.3, 92.5)  |
> | DGCNN  | 91.9 (91.9, 92.5)  |
> | CTNN   | **94.1 (94.0, 94.2)** |
>
> [1] Said, A., Bayrak, R., Derr, T., Shabbir, M., Moyer, D., Chang, C., & Koutsoukos, X. (2023). Neurograph: Benchmarks for graph machine learning in brain connectomics. Advances in Neural Information Processing Systems, 36, 6509-6531.

---

> ### Author Response · Authors · 2025-11-21
> **Responses to Reviewer vjKg Part 2**
>
> - **Adversarial graph structures.** A canonical example of an adversarial structure for our default labelers is a $d$-regular graph, where both degree and 1-WL assign identical labels to all vertices. More generally, highly symmetric graphs with large automorphism groups create many vertices with the same label and, consequently, many possible spanning trees; in this regime the cover is largely determined by random tie-breaking. For such graphs, stronger labelers such as closeness centrality or canonical labelers like NAUTY are more appropriate choices. In the revision, we have added a discussion of these adversarial structures, and how CTNN can be instantiated with stronger labelers to handle them.

---

> > ### Comment · Reviewer_vjKg · 2025-11-23
> >
> > Thanks for the effort. The authors have addressed most of my concerns, and I have raised my score accordingly.

---

> ### Author Response · Authors · 2025-12-02
>
> We thank the reviewer for their early engagement and are glad that the revisions have addressed most of their concerns! We appreciate the time taken in re-evaluating our work and are encouraged by the positive response.

---

### Official Review · Reviewer_ZKRx · 2025-10-27

**Soundness:** 3
**Presentation:** 3
**Contribution:** 2
**Rating:** 4
**Confidence:** 4

**Summary:**

This paper critiques single-sequence canonicalization methods in graph learning for causing distance distortion and having limited expressivity. To address this, it introduces Canonical Tree Cover Neural Networks (CTNNs), a framework that represents graphs using a small set of canonical spanning trees that cover all edges. Each tree is processed by a tree encoder, and the results are aggregated. The authors provide theoretical guarantees that CTNNs are probabilistically invariant, better preserve distances, and are more expressive than sequence-based methods. Empirically, CTNNs are shown to outperform standard GNNs, sampling approaches, and canonical sequence baselines on graph classification benchmarks.

**Strengths:**

1. The paper addresses a fundamental problem in graph learning: the trade-off between expressivity and isomorphism invariance.

2. It proposes a framework (CTNN) that uses a tree cover to represent graph structure, addressing the identified high distortion of sequence-based methods.

3. Theoretical analysis and empirical results across multiple benchmarks provide support.

**Weaknesses:**

1.  The experimental setup (e.g., using $\tau=1$) likely fails to meet the theoretical condition required by Lemma 5.3 for guaranteed logarithmic edge coverage. This gap between theory and practice undermines the paper's claims about efficient coverage and the universality (Thm 5.5) that depends on it.

2.  The empirical evaluation is missing the most critical baselines. As a method based on subgraph representations, CTNN must be compared against other state-of-the-art, subgraph-based GNNs (like GSN or ESAN) that also achieve high expressivity, not just standard 1-WL models.

3.  The universality claim (Thm 5.5) is trivial and not a unique advantage. This property holds for any model that completely decomposes a graph and uses universal encoders and aggregators. The paper does not sufficiently prove the *efficiency* of CTNN's universality (i.e., that it can be achieved with a small $K$).

**Questions:**

See in Weaknesses.

---

> ### Author Response · Authors · 2025-11-21
> **Responses to Reviewer ZKRx Part 1**
>
> We thank the reviewer for the encouraging feedback. We are pleased that the paper is seen as addressing a fundamental problem in graph learning between expressivity and isomorphism invariance, and that the tree-cover view of CTNNs is recognized as a way to reduce the high distortion of sequence-based canonicalizations. We also appreciate the acknowledgement that our theoretical and empirical results provide support for the framework.  Below, we address your specific comments individually.
>
> - **Do CTNNs obtain full coverage with $\tau=1$?** We agree that our theoretical condition in Lemma 5.3 may not be satisfied by the default choice $\tau = 1$ on *all* graphs. However, Lemma 5.3 provides a **sufficient** (not necessary) condition for logarithmic edge coverage. Smaller values of $\tau$ may yield high coverage even if they are not covered by the worst-case bound. Moreover, in Appendix E.1 (Fig. 6), we empirically verify that $\tau = 1$ achieves full edge coverage with only $K = 4$ trees on molecular datasets and $K = 8$ trees on protein datasets. For larger $\tau$ (e.g., 2, 4), full coverage is obtained even more rapidly (with smaller $K$), while both downstream performance and distortion remain unchanged across $\tau$ and $K$. This shows that in the regimes we study, CTNN obtains high coverage with few trees predicted by our analysis and is robust to the choice of $\tau$.
>
> - **Do CTNNs obtain universality efficiently with a small number of trees?** We agree that the universality theorem (Thm. 5.5) follows straightforwardly from full coverage and universal encoders and we include it primarily for completeness. The main theoretical novelty of CTNNs lies instead in (i) the coverage-aware spanning tree construction that yields a logarithmic-size tree cover under bounded arboricity (Lemma 5.3), and (ii) the accompanying distortion analysis, which shows that such covers preserve graph geometry substantially better than single-sequence canonicalizations.
>
>   Universality is explicitly conditional on the existence of a tree cover with full edge coverage; Lemma 5.3 provides this guarantee for sparse graph families. In this regime, CTNN attains universality with a provably small number of trees. Beyond the theoretical setting, we empirically verify that small $K$ and $\tau=1$ is sufficient on all benchmarks (Appendix E.1, Fig. 6).

---

> ### Author Response · Authors · 2025-11-21
> **Responses to Reviewer ZKRx Part 2**
>
> - **Comparison to expressive subgraph GNNs.** We agree that comparison to expressive subgraph-based GNNs beyond 1-WL MPNNs is important for evaluating CTNN. In the revised paper, we therefore add three such baselines, GIN+RWSE, GSN, and ESAN, on our molecular and protein benchmarks (Table 4) as well as on additional MoleculeNet classification benchmarks (Table 5). All three are strictly more expressive than 1-WL message passing and are representative of recent expressive subgraph/structural-encoding methods. Across the PCBA molecular datasets, CTNN achieves the best median AUC in all cases. On the protein benchmarks, CTNN attains the best median accuracy on PFAM and GO BIO and remains competitive on SCOP and GO MOL, where subgraph GNNs perform slightly better. On the additional MoleculeNet datasets in Table 5, CTNN achieves the best median AUC on 5 out of 6 tasks (all except SIDER), while remaining competitive on SIDER, where all approaches perform similarly. We hypothesize that although subgraph GNNs increase theoretical expressivity over MPNNs and hence improve their performance, they still fundamentally rely only on message passing and inherit known limitations such as oversmoothing and oversquashing. CTNNs, on the other hand, leverage powerful recurrent encoders on low-distortion spanning tree covers in tandem with message passing, allowing them to mitigate these limitations while retaining the benefits of MPNNs. These subgraph GNN baselines are included in the revised experimental section.
>
> **Table 4.** Median (min, max) performance of GIN, GIN+RWSE, GSN, ESAN, and CTNN on the molecular and protein benchmarks. CTNN achieves the best median score on all four PCBA datasets and on PFAM and GO BIO, while remaining competitive on SCOP and GO MOL.
> | Method      | PCBA-1030                | PCBA-1458                | PCBA-4467                | PCBA-5297                | SCOP                     | PFAM                     | GO BIO                   | GO MOL                   |
> |------------|---------------------------|---------------------------|---------------------------|---------------------------|---------------------------|---------------------------|---------------------------|---------------------------|
> | **GIN**    | 75.6 (71.3, 77.4)        | 85.7 (84.4, 86.4)        | 82.9 (81.8, 83.9)        | 92.2 (90.7, 92.5)        | 68.0 (67.9, 69.2)        | 20.0 (18.1, 21.0)        | 66.3 (59.9, 79.0)        | 83.7 (81.5, 85.6)        |
> | **GIN+RWSE** | 78.1 (76.9, 79.1)        | 87.9 (87.1, 89.5)        | 85.5 (82.9, 86.0)        | 92.5 (92.0, 94.3)        | 74.5 (72.1, 75.5)        | 17.6 (15.4, 21.0)        | 74.0 (69.8, 75.0)        | **85.8 (85.0, 86.1)**        |
> | **GSN**    | 76.9 (76.2, 77.3)        | 87.4 (86.4, 88.1)        | 83.1 (82.1, 83.4)        | 92.3 (91.9, 92.8)        | **74.5 (73.4, 76.7)**        | 15.1 (13.6, 16.5)        | 71.2 (59.0, 77.5)        | 85.0 (76.6, 85.3)        |
> | **ESAN**   | 74.7 (74.3, 75.0)        | 85.5 (85.3, 85.5)        | 80.3 (79.7, 81.6)        | 90.9 (90.7, 91.0)        | 66.6 (66.5, 68.5)        | 24.3 (19.0, 27.8)        | 74.8 (70.7, 75.7)        | 85.7 (85.6, 86.4)        |
> | **CTNN (ours)** | **80.6 (80.3, 81.2)** | **89.1 (88.0, 89.9)** | **86.8 (86.5, 87.4)** | **94.6 (94.2, 94.9)** | 72.0 (71.4, 72.3)        | **24.7 (20.9, 26.0)**    | **78.3 (77.9, 79.4)**    | 84.3 (84.0, 86.0)        |
>
> **Table 5.** Median (min, max) AUC of GIN, GIN+RWSE, GSN, ESAN, and CTNN on MoleculeNet benchmarks. CTNN achieves the best median AUC on 5/6 datasets (all except SIDER), while remaining competitive on SIDER.
>
> | Method       | CLINTOX                    | SIDER                      | BACE                       | TOXCAST                    | BBBP                       | TOX21                      |
> |-------------|----------------------------|----------------------------|----------------------------|----------------------------|----------------------------|----------------------------|
> | **GIN**     | 59.7 (54.1, 72.4)          | 66.5 (64.0, 69.9)          | 59.9 (51.4, 71.8)          | 55.7 (38.8, 60.8)          | 75.3 (49.4, 85.3)          | 66.9 (64.6, 73.4)          |
> | **GIN+RWSE**| 63.6 (56.4, 74.6)          | **66.7 (63.4, 71.3)**      | 57.7 (42.3, 69.6)          | 65.4 (53.3, 69.1)          | 73.7 (69.7, 86.5)          | 70.4 (66.1, 77.5)          |
> | **GSN**     | 63.7 (55.6, 68.2)          | 65.6 (62.4, 68.6)          | 70.1 (64.9, 79.1)          | 55.0 (49.3, 60.8)          | 71.4 (64.3, 79.7)          | 68.6 (60.7, 73.0)          |
> | **ESAN**    | 61.8 (56.7, 66.7)          | 65.9 (65.2, 70.8)          | 55.8 (52.3, 69.2)          | 65.8 (53.3, 69.5)          | 74.9 (70.5, 80.6)          | 70.3 (65.4, 75.9)          |
> | **CTNN (ours)** | **82.7 (56.8, 89.9)** | 64.1 (62.8, 67.3)          | **82.4 (79.7, 86.5)**      | **75.7 (70.0, 78.0)**      | **88.4 (83.7, 91.6)**      | **80.9 (79.6, 84.9)**      |

---

> > ### Comment · Reviewer_ZKRx · 2025-11-24
> >
> > The authors have strengthened the empirical evaluation by incorporating key baselines such as GSN and ESAN. While a minor gap remains between the theoretical assumptions and the experimental setup, the empirical results successfully demonstrate the robustness of the proposed method. I am satisfied with the authors' response.

---

> ### Author Response · Authors · 2025-12-02
>
> We thank the reviewer for their early and positive follow-up and for acknowledging the strengthened empirical evaluation! We are glad that the additional experiments helped clarify the practical robustness of CTNNs. We appreciate the reviewer’s constructive suggestions throughout the process and are encouraged by their satisfaction with the revised manuscript.

---

### Official Review · Reviewer_3g7j · 2025-10-27

**Soundness:** 2
**Presentation:** 3
**Contribution:** 2
**Rating:** 6
**Confidence:** 3

**Summary:**

The paper introduces Canonical Tree Cover Neural Networks (CTNNs), a new framework for graph representation learning that generalizes canonical sequence models by replacing a single canonical representation with a set of canonical spanning trees (MSTs).
Each tree is processed with a recurrent tree encoder, and message passing is applied over residual (non-tree) edges to capture local connectivity missed by individual trees.
The authors further provide:
(1) Theoretical results on probabilistic invariance and universality of CTNNs,
(2) Distortion and expressivity bounds comparing CTNNs to sequence-based canonicalization,
(3) Empirical evaluations on molecular and protein benchmarks, demonstrating improved performance.

**Strengths:**

(1) The paper presents a structured extension of canonical graph neural networks by introducing Canonical Tree Cover Neural Networks (CTNNs), which replace a single canonical ordering with a collection of spanning trees. The proposed method alleviates the structural distortion and expressivity limitations commonly observed in sequence-based canonicalization approaches.
(2) The paper provides a series of theoretical analyses, including probabilistic invariance, expected distortion bounds, and expressivity results, demonstrating a well-grounded theoretical foundation.

**Weaknesses:**

(1) Although the paper presents CTNN as an innovation grounded in canonicalization, its underlying modeling paradigm bears conceptual similarity to prior tree-structured graph neural networks (GNNs), such as Neural Trees for Learning on Graphs (Talak et al., 2021). Both approaches transform a graph into a hierarchy of trees for recursive message aggregation. As a result, the conceptual novelty of CTNN appears limited.
(2) Dependence on root and tree structure without theoretical guarantees. The model’s expressive capacity and stability appear highly sensitive to the root selection and tree shape. Since the recursive encoder (e.g., Tree-LSTM) is order- and hierarchy-dependent, different rootings or unbalanced spanning trees may yield substantially different representations. The paper currently provides no theoretical or empirical analysis of this effect.
(3) Dependency on the Initial Labeler ${\pi}_{V}$: The paper criticizes sequence methods (Prop 3.3) for being limited by ${\pi}_{V}$'s expressivity. However, CTNN's own tree generation (Alg. 1) is also initialized by ${\pi}_{V}$. If ${\pi}_{V}$ is weak (e.g., cannot distinguish 1-WL-equivalent nodes), the initial MST selection will also be "blind.
(4) Missing/Failed Key Baselines: A core claim is surpassing MPNN expressivity. A fair comparison requires stronger baselines like k-WL GNNs or Graph Transformers (GT). The paper includes GT, but it timed out (OOT) on all protein datasets where CTNN excelled. The lack of results from this key high-expressivity baseline makes CTNN's victory less convincing.

**Questions:**

(1) Considering the conceptual resemblance between CTNN and Neural Trees for Learning on Graphs (Talak et al., 2021), both of which convert graphs into hierarchical tree representations for recursive message aggregation, it would be valuable for the authors to include a direct conceptual or empirical comparison in the paper.
(2) Why were GNNs with proven higher expressivity than 1-WL (e.g., k-WL GNNs like GSN, or subgraph GNNs like PPGN) not included as baselines? These seem like the most relevant competitors for a method claiming to surpass 1-WL limitations.
(3) Given that the paper's core motivation is to overcome the expressivity limits of 1-WL, why were synthetic benchmarks (like CSL or EXP) completely avoided? These datasets are specifically designed to demonstrate 1-WL failures and would be the clearest way to validate the superior expressivity of CTNN.
(4) Given that recursive encoders (e.g., Tree-LSTM) are inherently order- and hierarchy-sensitive, how robust is CTNN to different root node selections or unbalanced tree shapes? Do the authors have any empirical results on performance variance under random root permutations or tree rebalancing?

---

> ### Author Response · Authors · 2025-11-21
> **Responses to Reviewer 3g7j Part 1**
>
> We thank the reviewer for the thoughtful and positive assessment. We are glad that the CTNN framework is seen as a meaningful extension of canonical graph neural networks that mitigates the distortion and expressivity limitations of sequence-based canonicalization. We are also encouraged that our probabilistic invariance, expected distortion bounds, and expressivity results, are viewed as providing a solid foundation for the method. Below, we address your specific comments individually.
>
> - **How do CTNNs differ from Neural Trees (Talak et al., 2021).** We appreciate the pointer to *Neural Trees for Learning on Graphs* (Talak et al., 2021) and agree that both methods exploit tree-structured computations. We will clarify the relationship in the revision. Conceptually, Neural Trees build an H-tree via repeated tree decompositions / junction trees, where each node of the H-tree corresponds to a *bag* (subgraph) of $G$; in contrast, CTNN constructs a *cover of spanning trees* in which every tree is an actual subgraph of $G$ whose nodes are in one-to-one correspondence with the original vertices. This structural difference is crucial: CTNN explicitly preserves graph distances along the trees, which lets us analyze coverage and distortion, whereas these notions are undefined for H-trees whose nodes are sets of vertices. Second, CTNN is designed around canonicalization and invariance: a node labeler and our MST-based construction yield a (probabilistic or deterministic, depending on the labeler) isomorphism-invariant tree cover, and any tree encoder plugged into CTNN inherits this invariance. Neural Trees do not rely on canonical node labelers and do not provide such invariance guarantees. Rather, their theory focuses on approximating G-compatible functions with complexity controlled by treewidth.
>
> - **Are CTNNs sensitive to root and tree structure?** We agree that CTNN’s performance depends on the choice of spanning trees and roots, which is why we design these components via canonicalization rather than leaving them random. To quantify this effect, we ran an additional ablation in which we replace CTNN’s coverage-aware, label-based tree construction with a uniformly random spanning tree with a random root on each tree (“RTNN” in Table 2). This variant uses exactly the same tree encoder as CTNN but removes all canonicalization. Across the molecular benchmarks, RTNN exhibits consistent drops in performance compared to CTNN, confirming that structured tree selection and root choice are crucial for performance. Intuitively, random trees can be highly unbalanced and rooted far from graph centers, forcing the recursive encoder to propagate information over long paths, whereas our canonical construction tends to produce trees that better respect the underlying graph geometry. We have added the RTNN ablation and accompanying discussion to make clear (i) that CTNN is intentionally designed to control root/tree dependence through canonicalization, and (ii) that removing this structure degrades performance in line with the reviewer’s intuition.
>
> **Table 2.** Median (min, max) AUC of RTNN (random trees) and CTNN with degree node labeler on MoleculeNet benchmarks (CLINTOX, SIDER, BACE, TOXCAST, BBBP, TOX21). CTNN consistently matches or outperforms RTNN, highlighting the benefit of canonical tree covers over random tree/root selection.
>
> | Method          | CLINTOX               | SIDER                 | BACE                  | TOXCAST               | BBBP                  | TOX21                 |
> |-----------------|-----------------------|-----------------------|-----------------------|-----------------------|-----------------------|-----------------------|
> | **RTNN**        | 76.8 (68.8, 85.1)     | **64.9 (61.8, 66.7)**     | 74.6 (68.1, 81.4)     | 70.7 (62.5, 76.2)     | 80.2 (70.1, 85.8)     | 76.8 (75.7, 79.2)     |
> | **CTNN**| **82.7 (56.8, 89.9)**     | 64.1 (62.8, 67.3)     | **82.4 (79.7, 86.5)**     | **75.7 (70.0, 78.0)**     | **88.4 (83.7, 91.6)**     | **80.9 (79.6, 84.9)**     |

---

> ### Author Response · Authors · 2025-11-21
> **Responses to Reviewer 3g7j Part 2**
>
> - **Are CTNNs limited in expressivity according to their initial node labeler?** We agree that if a model’s representation were solely determined by a weak node labeler $\pi_V$, its expressivity would be limited. However, CTNN is strictly more expressive than sequence-based canonicalization using the same labeler $\pi_V$. The key difference is that sequence methods apply a single $\pi_V$-defined ordering, so they cannot distinguish graphs that $\pi_V$ itself fails to distinguish. In contrast, CTNN uses $\pi_V$ only to *initialize* edge weights, while the evolving edge weights and multi-tree cover lead to full edge coverage and strictly more informative views of the graph. Lemma 5.4 formally shows that, for a fixed $\pi_V$, CTNN when equipped with the $\pi_V$ exceeds both (i) the expressivity of sequence-based canonicalization and (ii) the expressivity induced by $\pi_V$ alone. Our empirical results also reflect this result. We default to a weak, degree-based $\pi_V$ for scalability, yet CTNN still outperforms sequence canonicalization methods that use stronger labelers, supporting the claim that CTNN is not bottlenecked by the initial node labeler. Moreover, on CSL, a challenging isomorphism-testing benchmark, CTNN with a degree-based node labeler outperforms approaches with 1-WL and 2-WL expressivity and slightly surpasses a 3-WL–powerful architecture (PPGN/3WLGNN), further confirming that CTNN can go beyond the expressive limits of its initial node labeler (Table 3).
>
> - **Comparison to stronger expressivity baselines.** We agree that comparison to expressive subgraph-based GNNs beyond 1-WL MPNNs is important for evaluating CTNN. Our primary focus in the paper is on canonicalization baselines, in order to test whether CTNN improves over widely adopted canonicalization-based approaches for molecule and protein learning. In response to your suggestion, we have added three expressive subgraph/structural baselines, GIN+RWSE [1], GSN [2], and ESAN [3], on our molecular and protein benchmarks (Table 4 in response to Reviewer ZKRx) as well as on additional MoleculeNet classification benchmarks (Table 5 in response to Reviewer ZKRx). All three are strictly more expressive than 1-WL message passing and are representative of recent expressive subgraph/structural-encoding methods. Across the four PCBA datasets in Table 4, CTNN achieves the best median AUC in all cases. On the protein benchmarks, CTNN attains the best median accuracy on PFAM and GO BIO and remains competitive on SCOP and GO MOL, where subgraph GNNs perform slightly better. On the additional MoleculeNet datasets in Table 5, CTNN achieves the best median AUC on 5 out of 6 tasks (all except SIDER), while staying close to the strongest baselines on SIDER. We hypothesize that although subgraph GNNs increase theoretical expressivity over MPNNs and hence improve their performance, they still fundamentally rely on global message passing and inherit known limitations such as oversmoothing and oversquashing. CTNNs, on the other hand, leverage powerful recurrent encoders on low-distortion spanning tree covers in tandem with message passing, which helps mitigate these limitations while retaining the benefits of MPNNs. All of these additional subgraph GNN baselines are now included and discussed in the revised experimental section.
>
> [1] Dwivedi, V. P., Luu, A. T., Laurent, T., Bengio, Y., & Bresson, X. (2021). Graph neural networks with learnable structural and positional representations. International Conference on Learning Representations
>
> [2] Bouritsas, G., Frasca, F., Zafeiriou, S., & Bronstein, M. M. (2022). Improving graph neural network expressivity via subgraph isomorphism counting. IEEE Transactions on Pattern Analysis and Machine Intelligence, 45(1), 657-668.
>
> [3] Bevilacqua, B., Frasca, F., Lim, D., Srinivasan, B., Cai, C., Balamurugan, G., ... & Maron, H. (2021). Equivariant subgraph aggregation networks. International Conference on Learning Representations

---

> ### Author Response · Authors · 2025-11-25
> **Responses to Reviewer 3g7j Part 3**
>
> - **Inclusion of graph isomorphism testing benchmarks.** We agree that datasets such as CSL are useful for illustrating the limitations of 1-WL and for testing the expressivity of graph models. While our original focus was on molecular and protein benchmarks where canonicalization is widely used in practice and where we can study coverage and distortion at realistic scales, we agree that evaluating CTNNs on explicit isomorphism-testing benchmarks is valuable to directly assess expressivity. We have since conducted experiments on CSL [4], the circulant skip-link isomorphism testing dataset on which both 1-WL and 2-WL fail to distinguish any graphs (Table 3). We compare CTNN to MPNNs (1-WL expressive power), 2-IGN (2-WL expressive power) [5], and RingGNN [6] and 3WLGNN (PPGN) [7] (both 3-WL expressive power). As shown in Table 3. CTNN with $K=8$ trees using a degree node labeler significantly outperforms MPNNs, 2-IGN, and RingGNN, confirming that CTNNs can exceed the expressive power of 1-WL and 2-WL message-passing GNNs. Moreover, with $K=16$, CTNNs match and slightly surpass 3WLGNN performance. These additional results support our theoretical claims that CTNNs are strictly more expressive than MPNNs and can compete with or outperform architectures matching 2-WL and 3-WL expressivity. We will add the CSL experiments and discussion to the main paper.
>
> **Table 3.** Test accuracy (\%) on the CSL graph isomorphism testing benchmark. MPNNs, 2-IGN, and RingGNN (1–2-WL expressive power) perform equal to random, while CTNN with a degree node labeler achieves strong performance and matches/surpasses the 3-WL–expressive 3WLGNN when using more trees.
>
> | Approach        | CSL (ACC %)     |
> |-----------------|-----------------|
> | All MPNNs       | 10.0 ± 0.0      |
> | 2-IGN           | 10.0 ± 0.0      |
> | RingGNN         | 10.0 ± 0.0      |
> | 3WLGNN          | 95.7 ± 14.8     |
> | CTNN (K = 8)    | 88.6 ± 5.5      |
> | CTNN (K = 16)   | **96.6 ± 0.0**  |
>
> [4] Murphy, R., Srinivasan, B., Rao, V., & Ribeiro, B. (2019, May). Relational pooling for graph representations. In International Conference on Machine Learning (pp. 4663-4673). PMLR.
>
> [5] Maron, H., Fetaya, E., Segol, N., & Lipman, Y. (2019, May). On the universality of invariant networks. In International conference on machine learning (pp. 4363-4371). PMLR.
>
> [6] Chen, Z., Villar, S., Chen, L., & Bruna, J. (2019). On the equivalence between graph isomorphism testing and function approximation with gnns. Advances in neural information processing systems, 32.
>
> [7] Maron, H., Ben-Hamu, H., Serviansky, H., & Lipman, Y. (2019). Provably powerful graph networks. Advances in neural information processing systems, 32.

---

### Official Review · Reviewer_1KKi · 2025-10-28

**Soundness:** 2
**Presentation:** 3
**Contribution:** 3
**Rating:** 6
**Confidence:** 4

**Summary:**

This paper identifies limitations in existing graph canonicalization methods, particularly those that flatten graphs into sequences, arguing they introduce significant distance distortion and are bottlenecked by the expressivity of their node labelers. To address this, the authors propose Canonical Tree Cover Neural Networks (CTNNs), an invariant framework that represents each graph as a small set of canonical spanning trees. This tree cover is generated using an iterative, coverage-aware Minimum Spanning Tree (MST) algorithm. Each tree is then processed by an expressive tree encoder, and the results are aggregated. The authors provide theoretical contributions showing their method is (probabilistically) invariant, better preserves graph distances than sequence-based methods, and is strictly more expressive than both MPNNs and sequence canonicalization. Empirically, the paper demonstrates that CTNNs outperform standard GNNs, sampling approaches, and canonical sequence baselines on several graph classification benchmarks.

**Strengths:**

1. The paper is well-organized and clearly articulates the limitations of existing methods.

2. The proposed CTNN framework is a novel and intuitive solution that aims to tackle the issues by replacing the single-sequence representation with a more structurally tree cover.

3. The claims are supported by a combination of theoretical analysis and robust empirical results across diverse benchmarks.

**Weaknesses:**

1.  The term "canonicalization" is misleading. The method is not deterministic; it is a "probabilistically invariant sampling" algorithm that relies on random tie-breaking, making it conceptually closer to the sampling-based (e.g., RWNN) paradigms it critiques.
2.  There is a gap between the theoretical requirement for invariance (taking an expectation $\mathbb{E}[\cdot]$ over all possible random choices) and the practical implementation (aggregating a small sample of $K=4$ or $K=8$ trees). This small $K$ may be insufficient to approximate the expectation, leading to unstable representations for the same graph across different runs.
3.  A key theoretical justification (Thm 5.2) for the method's low distance distortion is based on Uniform Spanning Trees (USTs). However, the proposed algorithm (Alg 1) generates Minimum Spanning Trees (MSTs) from a different, non-uniform distribution, meaning the core distortion theory does not actually apply to the method as practiced.
4.  For highly regular graphs (e.g., where all nodes have the same degree $\pi_V$, resulting in identical initial edge weights ${\pi}_{E}^{(0)}$), the selection of the MST in Algorithm 1 will be determined entirely by the random tie-breaker $\zeta$. This suggests the process is not learning a "canonical" structure but rather performing a structured random decomposition of the graph.

**Questions:**

See in Weaknesses.

---

> ### Author Response · Authors · 2025-11-21
> **Response to Reviewer 1KKi Part 1**
>
> We thank the reviewer for the positive and detailed feedback. We are encouraged that the paper is found to be clear and well-organized, that the CTNN framework is viewed as a novel and intuitive way to move beyond single-sequence representations, and that our combination of theoretical analysis and empirical evaluation across diverse benchmarks is seen as supporting our claims. Below, we address your specific comments individually.
> - **Are CTNNs a “canonical” or random sampling framework?** We appreciate the reviewer’s observation and agree that our terminology should be clearer. CTNN is a *canonicalization-based framework* that is parameterized by a node labeler. When the labeler is a true canonical graph labeling that assigns unique labels for all nodes (e.g., the graph canonicalization tool NAUTY), the induced tree cover is a deterministic canonical representation: isomorphic graphs are mapped to exactly the same set of trees. In this regime, CTNN performs standard graph canonicalization.
>
>   In our experiments, however, we use weaker, structurally meaningful labelers (degree, centrality, 1-WL). These labelers are isomorphism-invariant but do not fully separate vertices, so we resolve ties at random, yielding probabilistic invariance: isomorphic graphs induce the same distribution over tree covers, which is sufficient to guarantee invariance in expectation. The trade-off is that canonical labelers like NAUTY, while eliminating randomness, often produce labels that are largely arbitrary from a modeling perspective (driven by lexicographic minimization rather than structural properties), whereas degree or centrality encode useful inductive biases such as hubs, periphery, and global importance. CTNN’s framework allows practitioners to choose along this spectrum: strictly deterministic but less structured labelings, or cheaper labelers with strong structural bias and probabilistic invariance. We have revised the invariance theorem to state this distinction explicitly: (i) a probabilistic invariance result for CTNN instantiations used in practice, and (ii) a deterministic invariance statement for CTNNs with canonical labelers.
>
>   Finally, although our current implementation is probabilistic, it differs from walk-based sampling methods: randomness appears only once during preprocessing, and the resulting low-distortion tree cover is reused throughout training, rather than continually re-sampling walks or searches. We have clarified this distinction and expanded the discussion of the above trade-off, including how one could instantiate CTNN with a deterministic canonical labeler if strict canonicalization is desired.
>
> - **Are CTNNs stable for small $K$? On regular graphs, can CTNNs provide meaningful canonical structures?** We agree that a small number of sampled trees $K$ may not perfectly approximate the expectation over all random tie-breaks for *all* graphs and node labelers, especially in adversarial cases such as highly symmetric $d$-regular graphs where a degree-based labeler assigns identical scores to all vertices and MST selection is driven entirely by random tie-breaking. In practice, however, we find that a small $K$ is sufficient on the real-world datasets we consider, where our existing labelers already break most symmetries. With the inexpensive degree labeler, increasing $K$ beyond 4 on molecular datasets and 8 on protein datasets does not improve edge coverage, downstream performance, or distortion (Appendix E, Fig. 6), indicating that the Monte Carlo approximation is already stable. Moreover, as the node labeler becomes stronger and separates more vertices (e.g., centrality or 1-WL), the number of distinct tie-breaking configurations, and thus possible tree covers, shrinks, so fewer trees are needed to approximate the expectation; correspondingly, we observe improved performance with these stronger labelers (Appendix E, Table 5). Finally, if one wishes to eliminate randomness altogether, CTNN can be instantiated with an off-the-shelf canonical graph labeler (e.g., NAUTY), in which case the induced tree cover is deterministic and strictly invariant rather than only invariant in expectation.

---

> ### Author Response · Authors · 2025-11-21
> **Response to Reviewer 1KKi Part 2**
>
> - **UST vs. CTNN Distortion.** We agree that Thm. 5.2 is stated for uniform spanning trees (USTs), whereas CTNN constructs coverage-aware MSTs from a different, non-uniform distribution. We use USTs primarily as a theoretically well-understood reference distribution to motivate why tree covers can achieve low distortion. Empirically, we verify that our coverage-aware MST scheme achieves even lower distortion than USTs on the datasets we consider. In Table 1, we compare the distortion for USTs and CTNN (degree labeler, $\tau=1$) on PCBA-1030 as $K$ increases. CTNN consistently attains a smaller average distortion than USTs for all $K$. We hypothesize that this advantage comes from the coverage-aware construction in Alg. 1, which explicitly favors edges that have not appeared in earlier trees, leading to tree covers that better preserve graph distances than independent UST samples, which have no bias toward uncovered edges. We have clarified in the paper that (i) Thm. 5.2 is used as a conceptual justification based on USTs, and (ii) our experiments provide quantitative evidence that the actual CTNN tree distribution inherits and can improve upon the low-distortion behavior established by the UST analysis.
>
> **Table 1.** Distortion as the number of trees, $K$, increases on PCBA-1030 for uniform spanning trees (UST) and CTNN, using a degree-based node labeler and $\tau = 1$.
>
> | Method | $K = 1$      | $K = 2$      | $K = 4$      | $K = 8$     |
> |--------|----------------|----------------|----------------|----------------|
> | UST    | 5.58 ± 1.23    | 4.49 ± 1.16    | 3.57 ± 0.70    | 2.92 ± 0.45    |
> | CTNN   | **5.10 ± 0.45**    | **3.08 ± 0.30**    | **2.26 ± 0.42**    | **2.13 ± 0.26**    |

---

> > ### Comment · Reviewer_1KKi · 2025-11-24
> >
> > I thank the authors for their detailed response. While concerns regarding theoretical purity—specifically the issue with regular graphs and the gap between UST theory and the MST implementation—persist, the additional experiments and clarifications provided have solidified the paper's contributions.

---

> ### Author Response · Authors · 2025-12-02
>
> We would like to thank the reviewer for their constructive feedback, as well as for engaging actively in the discussion phase. We are encouraged by the reviewers’ positive responses that our clarifications and additional experiments have solidified the paper's contributions.

---

### Author Response · Authors · 2025-12-02
**Overall Response**

We sincerely thank all reviewers for their time and effort into providing detailed feedback, which has improved our work.

We appreciate the reviewers highlighting the following strengths of our work:

- **Clear problem formulation and motivation (1KKi, vjKg).** Reviewers note that the paper is well-organized, clearly explains the limitations of existing methods, and uses intuitive examples to illustrate the shortcomings of sequence canonicalization.

- **Novel, intuitive CTNN framework (1KKi, 3g7j, ZKRx, vjKg).** Reviewers view CTNN as a structured and intuitive extension of canonical GNNs that replaces a single canonical ordering with a collection of spanning trees, directly addressing the high distortion and expressivity limitations of sequence-based canonicalization.

- **Strong theoretical foundation (3g7j, ZKRx, vjKg).** Reviewers emphasize that the probabilistic invariance analysis, expected distortion bounds, coverage guarantees, and expressivity results provide a well-grounded theoretical basis for CTNNs.

- **Robust empirical support (1KKi, vjKg, ZKRx).** Reviewers highlight that CTNNs consistently surpass strong baselines, including expressive subgraph methods, achieving notable gains on molecular and protein benchmarks.

During the discussion phase, we responded to all reviewer suggestions and criticisms. **After our revisions, reviewers 1KKi, ZKRx, and vjKg each expressed satisfaction with our responses and were positive about the work, while reviewer 3g7j was positive prior to the discussion.** We summarize our updates below.

- **Canonicalization vs. Probabilistic Invariance and adversarial structures.** We clarified that CTNNs are a canonicalization-based framework that interpolates between deterministic canonicalization and probabilistic invariance, depending on the choice of node labeler (1KKi). For adversarial structures such as regular graphs, we now include results on the challenging CSL isomorphism benchmark (Table 3), where CTNN with a weak degree labeler outperforms 1-WL, 2-WL, and 3-WL approaches, demonstrating its ability to distinguish difficult regular graphs (1KKi, 3g7j, vjKg).

- **Comparison to expressive subgraph GNNs.** We now include results on three expressive subgraph/structural baselines: GIN+RWSE, GSN, and ESAN, each more expressive than 1-WL and representative of recent subgraph approaches (Tables 4 and 5; ZKRx, 3g7j). Across all benchmarks, CTNN consistently outperforms or matches these subgraph GNNs.

- **Clarification and empirical validation of theoretical results.** (i) While our distortion analysis is formulated for uniform spanning tree (UST) distributions as a well-understood reference, we empirically verify that CTNN’s coverage-aware minimum spanning tree distribution inherits and can even improve upon the low-distortion behavior of USTs (Table 1; 1KKi). (ii) We emphasize that CTNNs provably exceed the expressivity of their node labeler (Lemma 5.4; 3g7j), a fact that is reflected empirically: CTNNs equipped with a cheap degree labeler obtain the best performance among canonicalization methods. On the challenging CSL benchmark, CTNN with this inexpensive degree labeler outperforms 1-WL, 2-WL, and the 3-WL–powerful GNNs (Table 3), further confirming that CTNN is not bottlenecked by the initial node labeler. (iii) In addition, we verify that CTNN achieves full edge coverage for small $K$ across choices of $\tau$ (Appendix E.1), and that universality is provably and efficiently realized with a logarithmic number of trees (ZKRx).

- **CTNN runtime and memory.** We report quantitative preprocessing times (Appendix E.2; Table 6; vjKg), showing that constructing canonical spanning trees is fast across all datasets, including larger brain graphs. The preprocessing step is parallelizable across graphs, performed once per dataset, and adds only an $O(Kn)$ memory for storing the tree covers, demonstrating that CTNNs are scalable and efficient in practice.

- **Evaluation beyond biochemical datasets.** We additionally evaluate CTNN on a larger brain graph classification benchmark from NeuroGraph, where CTNN outperforms baselines (Table 7; vjKg). This demonstrates that CTNN’s benefits extend beyond molecular and protein graphs to domains where canonicalization is not yet widely adopted.

---

### Meta-Review · Area_Chair_zrSm · 2026-01-07

**Summary:**

The paper introduces Canonical Tree Cover Neural Networks (CTNNs), a new framework for learning invariant graph representations. Instead of flattening a graph to a single canonical sequence, CTNN builds a small set of canonical spanning‑tree covers that together cover all edges; each tree is processed by an expressive tree encoder (a bidirectional Tree‑LSTM) and optionally a residual GIN‑based message‑passing layer on the non‑tree edges. The per‑tree embeddings are summed to obtain a permutation‑invariant graph embedding. The authors prove that CTNNs are strictly more expressive than any sequence‑based canonicalization pipeline, preserve graph distances with lower stretch and contraction, and achieve universality on invariant graph functions when equipped with a universal tree encoder and sufficient coverage (logarithmic‑size covers for sparse graphs). Empirically, CTNN consistently outperforms invariant GNNs, random‑walk‑based samplers, and all evaluated sequence‑based canonicalizations across eight graph‑classification benchmarks (molecular PCBA tasks, protein classification, and a brain‑graph benchmark), showing higher AUC/accuracy and lower distortion while incurring modest preprocessing cost.

**Reviewer Concerns:**

- Reviewers ZKRx and vjKg view the contribution as somewhat incremental and point to remaining gaps (e.g., distortion analysis for the MST‑based cover, reliance on weak labelers, and lack of stronger expressive baselines).
- Reviewer ZKRx noted that the chosen $\tau = 1$ might violate a lemma’s sufficient condition and that the universality claim lacks discussion of efficiency.
- Reviewer 1KKi questioned the terminology of “canonicalization,” argued that using only a few trees (K = 4 or 8) might give unstable representations, noted that the distortion analysis was proved for uniform spanning trees rather than the MSTs actually used, and pointed out that for highly regular graphs the method relies heavily on random tie‑breakers.
- Reviewer 3g7j highlighted the similarity to Neural Trees, asked for analysis of sensitivity to root and tree shape, criticized the dependence on weak initial labelers, and noted the absence of stronger expressive baselines and synthetic benchmarks that expose 1‑WL failures.
- Reviewer vjKg considered the contribution incremental, observed that the theoretical guarantees (logarithmic tree cover, low distortion) mainly apply to sparse graphs, felt the empirical gains were modest, and noted missing runtime/memory analysis and evaluations on larger non‑biochemical graphs.

**Reviewer Scores:**

- All reviewers that replied are satisfied with the authors’ revisions and consider the contributions to be solid. In particular, these reviewers say that the empirical evaluation is now stronger (additional baselines, distortion tables, runtime data).
- Reviewers acknowledge that the paper presents a novel framework (CTNN) that improves expressive power and reduces distance distortion compared with existing canonical‑sequence and 1‑WL‑based methods.
- I am generally not a fan of canonicalization (brittle) but this approach may have enough diversity in the trees to not face as many issues in practice.

In the rebuttal the points addressed:
- Authors clarified that CTNN can be deterministic when a true canonical node labeler (e.g., NAUTY) is used, and probabilistic with weaker labelers.
- Authors showed that the chosen numbers of trees (K = 4 for molecules and K = 8 for proteins) achieve full edge coverage and stable performance, and provided Fig 6 (Appendix) as evidence.
- Authors acknowledged that the distortion theorem is proved for uniform spanning trees (UST) and added empirical Table 1 demonstrating that CTNN’s distortion is ≤ UST for all values of K.
- Authors added an explicit conceptual comparison to Neural Trees (Talak et al., 2021), emphasizing that CTNN’s trees are actual spanning subgraphs that preserve distances and that CTNN guarantees isomorphism invariance via canonical labelers. Authors also included three expressive‑subgraph baselines (GIN+RWSE, GSN, ESAN) in Tables 4 and 5, showing that CTNN matches or exceeds them on most tasks.
- Authors also added the CSL isomorphism benchmark (Table 3) to demonstrate superiority over 1‑WL and 2‑WL models and competitiveness with 3‑WL models.
- Authors clarified that the chosen $\tau = 1$ works well in practice; Fig 6 shows full edge coverage with the reported K values, and the lemma’s condition is sufficient but not necessary.
- Authors also discussed the universality claim, explaining that it follows from full edge coverage together with a universal tree encoder, and noted that efficiency is ensured by the logarithmic‑in‑size tree cover guaranteed for bounded‑arboricity graphs (Lemma 5.3).
- Authors also provided quantitative preprocessing runtime (Table 6) and memory overhead ( O(K n) ), demonstrating the method’s practical efficiency.

---

### Decision · Program_Chairs · 2026-01-26

Accept (Poster)